# Innovative Photonic Sensors for Safety and Security, Part III: Environment, Agriculture and Soil Monitoring

**DOI:** 10.3390/s23063187

**Published:** 2023-03-16

**Authors:** Giovanni Breglio, Romeo Bernini, Gaia Maria Berruti, Francesco Antonio Bruno, Salvatore Buontempo, Stefania Campopiano, Ester Catalano, Marco Consales, Agnese Coscetta, Antonello Cutolo, Maria Alessandra Cutolo, Pasquale Di Palma, Flavio Esposito, Francesco Fienga, Michele Giordano, Antonio Iele, Agostino Iadicicco, Andrea Irace, Mohammed Janneh, Armando Laudati, Marco Leone, Luca Maresca, Vincenzo Romano Marrazzo, Aldo Minardo, Marco Pisco, Giuseppe Quero, Michele Riccio, Anubhav Srivastava, Patrizio Vaiano, Luigi Zeni, Andrea Cusano

**Affiliations:** 1Dipartimento di Ingegneria Elettrica e delle Tecnologie dell’Informazione, Università degli Studi di Napoli Federico II, Via Claudio 21, 80125 Napoli, Italy; 2European Organization for Nuclear Research (CERN), 1211 Geneva, Switzerland; 3Istituto per il Rilevamento Elettromagnetico dell’Ambiente, Consiglio Nazionale delle Ricerche, Via Diocleziano 328, 81024 Napoli, Italy; 4Gruppo di Optoelettronica e Fotonica, Dipartimento di Ingegneria, Università degli Studi del Sannio, Corso Garibaldi 107, 82100 Benevento, Italy; 5National Institute for Nuclear Physics (INFN), 80125 Napoli, Italy; 6Dipartimento di Ingegneria, Università Degli Studi di Napoli Parthenope, Centro Direzionale Isola C4, 80143 Napoli, Italy; 7Dipartimento di Ingegneria, Università della Campania Luigi Vanvitelli, Via Roma 29, 81031 Aversa, Italy; 8Optosensing Ltd., Via Carlo de Marco 69, 80137 Napoli, Italy; 9Istituto per i Polimeri, Compositi e Biomateriali Consiglio Nazionale delle Ricerche, Via Enrico Fermi 1, 80055 Portici, Italy; 10CERICT SCARL, CNOS Center, Viale Traiano, Palazzo ex Poste, 82100 Benevento, Italy; 11Optosmart Ltd., Via Pontano 61, 80122 Napoli, Italy

**Keywords:** optical fiber sensors, fiber Bragg gratings, long period gratings, distributed sensing, soil monitoring, radiation sensors

## Abstract

In order to complete this set of three companion papers, in this last, we focus our attention on environmental monitoring by taking advantage of photonic technologies. After reporting on some configurations useful for high precision agriculture, we explore the problems connected with soil water content measurement and landslide early warning. Then, we concentrate on a new generation of seismic sensors useful in both terrestrial and under water contests. Finally, we discuss a number of optical fiber sensors for use in radiation environments.

## 1. Introduction

The sensing and monitoring application fields have highly benefitted from photonic and optical fiber technologies. Our group, consisting of researchers with different scientific backgrounds and from different universities in Campania, Italy, has been engaged in the development of photonics and nanophotonic sensing devices for the last twenty years. The domains where we have applied such sensors are several, including both medical and industrial applications (e.g., CERN, railways and aerospace) [1,2,3,4,5,6,7,8,9,10,11,12,13,14,15,16,17,18,19,20,21,22,23,24,25,26,27,28,29,30,31,32,33,34,35,36,37,38,39,40,41,42,43,44,45,46,47]. Therefore, we collected in three companion papers (to which this paper belongs) our main results regarding photonic sensors for safety and security. This paper is the third (Part III) and reports on the applications together with Part II [48], while theoretical background of the applied technologies can be found in Part I [49].

This work reports on novel photonic sensor applications in agriculture, in seismic monitoring, and in environments exposed to radiation. Specifically, soil monitoring is first presented by focusing on the measurement of water content and on landslide early warning. Afterward, seismic monitoring is reported by considering both land and underwater environments. Finally, a number of applications related to high-energy physics experiments at CERN involving the presence of radiations are illustrated.

## 2. Optical Fiber Sensors for Agriculture and Soil Monitoring

In the next sections, the investigations carried out by our multidisciplinary research group concerning the development of fiber optic sensors for water content measurements in soil and landslide early warning are presented.

### 2.1. Optical Fiber Sensors for Soil Water Content Measurement

Real time measurements of soil water content play a critical role in many fields of science such as agronomy, geology, engineering and hydrology. In recent years, soil water sensor technology has been increasingly employed in agriculture to optimize the irrigation process and guarantee a sustainable water resources management.

Recent technological innovations introduced by environmental monitoring systems have played a crucial role in the development of precision agriculture and appear to be a major factor in the sustainable expansion of agricultural systems. Many sensing applications have a significant impact on farming practices. For example, soil moisture sensors may assist farmers in making irrigation decisions in order to protect crops from drought stress and excessive irrigation [50]. Various types of soil water sensors are available in the market (e.g., capacitive, resistive, dielectric and tensiometric sensors) [51], but they are not suitable for monitoring large areas. In contrast, fiber optic sensors (FOSs) are recommended for multipoint monitoring of large areas due to their multiplexing capability and wavelength-encoded information property, thus allowing reduction of the final system cabling complexity with the use of single compact interrogation system [52]. In 2014, our multidisciplinary research group was involved in the development of a new generation of soil moisture and temperature sensors based on Fiber Bragg Grating (FBG) technology for irrigation purposes [53]. The developed solution consisted of two FBGs, one coated with a sensitive polyimide layer for relative humidity measurement (RH), while the other one was free for temperature measurement and thermal compensation [54,55,56,57], as schematically shown in Figure 1a.

The final device was integrated in a customized aluminum protection package (see picture in Figure 1b), designed ad hoc, with a polymer micro-porous membrane. The role of such membrane was to prevent the FBGs from being in direct contact with water during the liquid phase, while allowing water in the gaseous phase to pass through and interact with the FBG coating.

The proposed fiber optic thermo-hygrometer was calibrated against RH and temperature in an industrial climatic chamber. Afterward, an extensive experimental campaign was carried out by embedding the realized probe in several soil samples prepared by using the gravimetric method [58] at different volumetric water content (VWC) values, and experimental tests were performed in order to find out the relationship between the RH value provided by the FBG thermo-hygrometer and the soil VWC.

Figure 2a reports the response of the proposed thermo-hygrometer when it is buried in several soil samples with different VWC values (from 0% to 20%). As shown, the relative humidity drops to values close to 5% when the device is buried in dry soil (VWC = 0%), while it suddenly increases over 90% at VWC values above 5%. The sensitivity of the device, evaluated as the first derivative of the fitting curve shown in Figure 2b, was found to be significant in the range [0–5]% VWC, while a saturation effect was observed at VWC values above 10%.

To overcome the limitations of the proposed device in terms of measurement range, in 2017 we experimentally demonstrated a novel solution for soil moisture content monitoring based on the integration of an FBG sensor into a larger custom-made polyvinyl chloride (PVC) cylindrical package [53,58], as schematically illustrated in Figure 3. The upper part of the cylinder, sealed with a hermetic plug, interacts with the bottom through a microporous hydrophobic membrane covering the lower part. When buried in the soil, a certain amount of water molecules in the vapor state passes over thus being distributed throughout the package volume in function of the soil VWC. In particular, for a given VWC value, the concentration of the water molecules within the package (and thus the RH) is closely related to its geometrical characteristics (e.g., diameter, volume, height, etc.).

To validate such measurement principle, the thermo-hygrometer was inserted into a PVC functional package with an outer diameter of 2.2 cm and a total volume of 300 cm^3^. A preliminary test was performed by burying the optical device and a reference VWC sensor in a soil sample at ~4% VWC. The soil was gradually irrigated in order to reach three increased VWC levels. Collected results, reported in Figure 4, confirmed the capability of the “functional” package to extend the VWC measurement range of the fiber optic soil moisture sensor and paved the way for a more in-depth investigation.

The optimized soil moisture sensor calibration curve (RH versus VWC) is illustrated in Figure 5. Collected results confirmed that the proposed device is able to provide measurements from 0% VWC—corresponding to dry soil—up to 37% VWC. A piecewise linear fitting was applied to the data set reported in Figure 5. As evident, the device exhibits a high sensitivity in the order of 5.4%RH/%VWC in the low VWC range ([0–14]%), while it drops about 0.3% RH/%VWC at higher VWC values, due to the saturation effect.

In order to maintain the advantages of fiber optic-based technology for soil water content measurement applications, we presented in 2021 an innovative and cost-effective solution based on the combination of a functional material consisting of a nanoporous ceramic disk and an engineered optical fiber probe operating in the NIR range [59]. The working principle of the proposed device is based on the VWC sensitivity of the Al_2_O_3_ absorbance spectra in the NIR [60,61]. A feasibility analysis was conducted in order to demonstrate the capability of the proposed functional material to absorb and release water and to confirm the possibility of measuring the optical properties changes in the NIR region. To this aim, an experimental campaign was performed to verify the ceramic disk water absorption capability directly into soil. Finally, a compact sensor prototype, consisting of a Y-shaped bifurcated cable (see Figure 6a) coupled to a nanoporous ceramic disk, was designed to reduce the cost and the complexity of the whole platform and make it directly applicable in real scenarios.

The bifurcated Y-shaped cable was designed in order to accommodate two fibers in a single body. Specifically, the fibers are juxtaposed at the common end, converge in an SMA905 connector, and split into two legs, one for the light source connection and one for the detector, both terminating in an FC/PC connector. The common leg illuminates and collects light from the disk. A 6 mm thick stainless-steel tube was used to protect the fibers, with a short coaxial plastic sheath at both ends to make the optical cable more robust and flexible at the same time. In addition, a suitable structure made of some optical commercial components was used in order to couple the optical fiber (common leg side) together with the nanoporous ceramic disk, as reported in Figure 6b.

The test was carried out by burying the realized device inside dry soil and gradually irrigating it to obtain three increasing VWC steps, each one of about 10%. The soil water content was monitored by using a frequency-domain reflectometry (FDR)-based commercial reference sensor.

Collected results, reported in Figure 7a, show the evolution of the reflected power spectrum during the experimental test. As expected, the power decreases as soon as the prototype is buried in the soil, while the reflected power spectrum decreases when the water content increases. The first calibration curve (ΔP vs. VWC) was evaluated, as shown in Figure 7b. By fitting the ΔP values with a linear curve in the range from 3 to 35% VWC, it was found that the device is characterized by a high sensitivity of about −2.3% VWC in a wide VWC range, achieving a resolution lower than 1% VWC.

### 2.2. Landslides Early Warning

Landslides are caused by disturbances in the natural stability of a slope. To reduce the risk of landslide, early-warning systems can be adopted by which the triggering causes or their consequences, such as displacement of soil and acceleration, are supervised in real-time. In this context, distributed optical fiber sensors offer a powerful tool as they permit the monitoring of the slope movements by fixing the fiber to the soil or using geogrids/geotextiles [62,63,64].

In this section, we report two examples of application of this technology to the geotechnical monitoring field: the first one is an experiment carried out in a laboratory scenario with an instrumented flume, while the second one is the result of a long-term monitoring campaign carried out in a field scenario. For the flume experiment, a high spatial resolution (5 cm) distributed optical fiber sensor based on the Brillouin Optical Frequency-Domain Analysis (BOFDA) was chosen. The purpose of this experiment was investigating the failure of a small-scale volcanic slope subjected to artificial rainfall [65]. For our experiments, soil from the Cervinara site was employed (northeast of Naples, Italy), an area affected by a devastating debris avalanche in 1999. The soil was laid down by placing 1 cm thick layers, which were gently tamped while wetted, up to a total thickness of 10 cm. The length and width of the slope were 110 cm and 50 cm, respectively. In the described experiment, the imposed slope angle was 35°, therefore much lower than the friction angle of the soil (38°). A standard single-mode optical fiber was installed into the flume at a depth of 5 cm to measure the soil strain field.

The optical fiber was laid inside the instrumented flume along two transversal sections (OF-S1 and OF-S2 in Figure 8a) and two longitudinal sections (OF-S3 and OF-S4 in the same figure). A slight pretension was applied to the installed fiber, while fixing the latter to the walls of the flume by glue.

The optical fiber measurements are illustrated in Figure 8b, where the measured strain is reported as a function of the curvilinear abscissa along the fiber. The figure clearly reveals the peaks of strain related to the pretensioned strands embedded into the slope (OF-S1, OF-S2, OF-S3 and OF-S4). The measurements also reveal the progressive soil deformation consequent to the artificial rainfall of 100 mm/h reproduced above the ground surface since the beginning of the experiment. Figure 9 compares the strains measured by the fiber sections OF-S1 and OF-S2, with those calculated from the images acquired by a digital camera implementing the Particle Image Velocimetry (PIV) technique. The two techniques provide similar strain during the first part of the experiment while, after the opening of the tension crack, only the optical fiber sensor was able to track the large strains occurring in the last 10 min before the slope failure. Moreover, the fiber strand OF-S2, located close to the cracked area, was able to detect a higher soil strain with respect to the PIV data at the same position (see Section 2 in Figure 9) and early (about 4 min) during the experiment. This suggests that the optical fiber sensor can be used not only to evaluate the slope strain field during a rainwater infiltration process but also to realize early warning through the detection of the precursor signals of an incoming failure.

The second example is a long-term monitoring campaign carried out in a railway tunnel located in a landslide prone area [66]. For these tests, a portable Brillouin Optical Time-Domain Analysis (BOTDA) sensor unit was used, featuring a spatial resolution of 1 m. The sensor was employed to detect the strain distribution along the two sidewalls of a 200 m long railway tunnel (tunnel “Calabrese”), consisting of eight contiguous sectors separated by joints with irregular spacing. As the sensing element, a single-mode 0.9 mm tight-buffered optical fiber was hand-glued along the two sidewalls of the tunnel using epoxy adhesive. Figure 10 reports the results relative to the whole monitoring campaign, separately shown for the upslope and downslope tunnel sidewalls, respectively. The vertical dashed lines indicate the position of the structural joints. Several peaks are visible in the acquired strain profiles. In some cases, these peaks are located in the position of the structural joints. Through careful inspection of the tunnel conditions, we have identified four different phenomena which may give rise to the observed peaks. These are indicated by a bracketed letter in Figure 10.

As seen in Figure 10, the letter J indicates the occurrence of a peak in correspondence to a structural joint of the tunnel. Instead, the letters C, PS and SP correspond, respectively, to a crack, a local parget swelling on the tunnel walls and a salt precipitate accumulation due to water infiltration and evaporation, exemplified in the pictures shown in Figure 11. As apparent from the strain data reported in Figure 10, the highest deformations are to be attributed to the local swelling of the parget. However, from the point of view of the monitoring of the landslide, more meaningful information can be extracted from the peaks occurring in correspondence to the joints. More specifically, the highest strains are observed at joints 1, 2 and 3. It is important to point out that those joints were also the ones where the higher displacements were measured by a caliper since the tunnel construction in 1992. Another feature that can be retrieved from the strain measurements reported in Figure 10 is that the deformation recorded along the downslope wall is higher than that recorded along the upslope wall. This fact agrees with the local kinematics of the landslide accumulation, which opens similarly to a fan toward the river, that, in turn, activates the movements by erosion.

The fiber elongation in correspondence of the joint’s opening was determined following the procedure described in Ref. [67]. In brief, the strain values around each J peak were detrended and integrated along a fiber length corresponding to the spatial resolution of 1 m. Figure 12 shows the time series of the computed fiber elongation along the upslope and downslope tunnel walls, respectively.

As apparent from Figure 12, significant elongations are recorded in correspondence to joints 1, 2 and 3 starting from the second half of 2016. This is consistent with the fact that in that area a small landslide was triggered by the river erosion. Starting from June 2018, i.e., about two years after the fiber installation, a significant elongation is observed in correspondence to all the joints, while the fiber even broke in correspondence of the joints 1 and 2 along the downslope wall. This trend agrees with the displacements obtained from GPS measurements [68]. It is finally worth to note that according to a recent analysis of Cosmo-SkyMed data [69], an acceleration of the movements in the accumulation area occurred after 2018, which may explain the breakage of the fiber corresponding to joints 1 and 2. The experimental results thus demonstrate that the monitoring system based on distributed optical fiber strain sensing is a reliable and effective methodology to monitor the tunnel behavior.

## 3. Optical Fiber Sensors for Seismic Monitoring

Every day, several seismic events with different magnitude occur worldwide both on the mainland and underwater sea. The increasing demand of capillary and high-performance monitoring stations all over the world continuously requires technological advancements in the sensing devices and monitoring systems. In the next sections, we describe our recent research activities devoted to the development of optical fiber sensing systems for geophysical and volcanological monitoring [70,71].

### 3.1. Optical Fiber Sensors for Land Seismic Monitoring

An earthquake consists of a series of natural vibrations in the ground and is caused by the sudden release of energy within the lithosphere. When rocks break, they release the accumulated energy in the form of elastic waves, which reach the surface causing ground shaking and propagating also over long distances. Consequently, such waves, if of strong intensity, can lead to catastrophic events causing damage to buildings and the entire population [72]. Statistically, the earthquakes with greater intensity and energy occur at the limits of the tectonic plates, i.e., along the margins of the Pacific Ocean and along the Alpine–Apennine–Himalayan belt [73].

The Italian national territory is characterized by a high seismic risk [74]. In Italy, in the years 2010 and 2020, the Italian National Institute of Geophysics and Volcanology (INGV) recorded over 350 earthquakes with a magnitude greater than 4.0 and 35 greater than 5.0. In areas with high seismic risk, the damage that earthquakes can cause is significant both on buildings and on citizens’ lives. Therefore, it is important to develop and use increasingly efficient and innovative monitoring systems to mitigate risks and safeguard society by increasing safety levels. The use of earthquake monitoring systems allows for detecting and identifying the magnitude and epicenter of an earthquake but enables also further studies on underlying seismic phenomenology [75].

Seismic monitoring techniques on the mainland take place through the use of seismometers, accelerometers and geophones. These technologies have high performance in terms of bandwidth, sensitivity and dynamic range. However, it should be specified that they have disadvantages associated with their practical use, especially when a high number of measurement points is required or when the installation must be performed in open spaces [76].

Fiber optic sensors represent a valid alternative to the aforementioned conventional technologies as they are light, not bulky, and immune to electromagnetic interference. Additionally, they are suitable to be used in large environments, as they have multiplexing capabilities and allow remote operation, resulting in simpler installations and less complex wiring than conventional technologies [41,77,78,79].

A few optical fiber seismic sensors were proposed in the literature. In many investigations, FBG sensors were integrated with various mechanical structures to obtain a response to seismic waves. Nonetheless, these configurations have lower sensitivities than conventional technologies [80,81,82,83,84]. Recently, we proposed for the first time the integration of optomechanical cavities with the optical fiber to obtain miniaturized devices suitable to detect earthquakes with performance compatible with conventional technologies and therefore capable of detecting even small ground movements. This approach, involving the integration of nano- and micro-opto-mechanical systems [85] with optical fiber by using Lab on Fiber technology [86,87], was successfully adopted also in other scenarios. To date, for example, Digonnet et al. [88,89] proposed a hydrophone based on a photonic-crystal diaphragm positioned on the fiber tip. Iannuzzi et al. developed a nanoindenter instrument [90] by fabricating micro-cantilevers on the optical fiber tip [91,92,93]. In the following, we report on the development and field test of a seismic accelerometer relying on an optomechanical cavity constructed on the fiber tip.

### 3.2. Opto-Mechanical Lab on Fiber Seismic Accelerometer

The innovative optomechanical sensor is constituted of a dual-beam cantilever with a properly designed proof mass suspended on the optical fiber tip. The schematic of the sensor is shown in Figure 13a. Specifically, the mechanical structure consists of an X-shaped glass cantilever beam with the crossing point suspended in correspondence to the optical fiber end facet. A small additional mass is placed on the cantilevers free end, while the other cantilever ends are clamped on the sensor body. The principle of operation is schematically shown in Figure 13b. When a ground vibration occurs, the cavity length changes and consequently changes the optical path of the light between the cantilever free end and the fiber end facet. The cavity path variation can be optically detected using interferometric interrogation techniques [70]. We performed full 3D numerical simulations in order to carefully tailor the size of the opto-mechanical structure and therefore optimize the sensitivity and bandwidth of the seismic sensor. Indeed, as we have shown in a previous work [70], the characteristics of the micromechanical structure integrated on the fiber tip can be used to tune the sensitivity and bandwidth of the accelerometers. In Figure 13c, we display, as an example, the deformed shape representation of the total displacement when the structure is subject to a vertical acceleration at 5 Hz, while in Figure 13d we report the responsivity of one of the designed sensors, expressed in terms of cavity length variation with respect to the vertical acceleration [70].

The designed sensor was fabricated by using a ferrule top approach [70]. The fabricated sensor was preliminarily characterized in the lab. In detail, the dynamic response of the developed prototype was obtained by comparison with the response of a piezoelectric accelerometer in the presence of an impulse vibration induced by a hammer blow. In Figure 14a,b, we show the resulting amplitude and phase of the sensor responsivity. The responsivity features an in-band responsivity of 1.5 µm/(m/s^2^), a 3 dB bandwidth of approximately 60 Hz and a resolution for detecting seismic waves of 0.44 µg/√Hz. Then, the characterization was extended down to 0.1 Hz using the “ground as a shaking table”, that is, by using natural sources of vibration at low frequencies (see Figure 14c).

Successively, the Lab on Fiber seismic accelerometers were integrated in a conventional seismic station in Naples, Italy, and used for seismic surveillance applications. During the field trial, a sequence of seismic events occurred throughout central Italy. The lab-on-fiber seismic accelerometer (positioned at 300 km from Napoli) clearly recorded the seismic swarm (Figure 15a). In particular, a Mw6.5 earthquake struck Norcia, in Italy, on 30 October 2016. In Figure 15b we show the comparison between the traces of the reference sensor (Episensor FBA-EST Kinemetrics) and our optical fiber sensor in term of displacement. The trace overlap clearly highlights the capability of the optical sensing system to operate as a seismic accelerometer.

### 3.3. Fiber Optic-Based Sensors for Underwater Seismic Monitoring

Currently, most seismic stations are located on the mainland. Nonetheless, most of the earth’s surface is covered with water, and the need for seismic monitoring to retrieve key information on tectonic movements is not limited to the mainland. Additionally, the sudden underwater displacement of tectonic plates can be transmitted to the overlying water column, resulting in the formation of a dangerous wave on the surface, the tsunami. Therefore, high performance underwater acoustic detection equipment is urgently required to provide widespread monitoring systems, in which hydrophones are considered as one of the core components [94]. For example, in southern Italy, west of the city of Napoli is located the Campi Flegrei caldera, which continuously exhibits intense activity and extends from the land to the sea.

INGV has developed a marine research infrastructure called MEDUSA, Multiparametric Elastic-beacon Devices and Underwater Sensors Acquisition system, in order to monitor local volcanic activity for obtaining important data on the earthquakes phenomenology. Medusa consists of geodetic buoys equipped with instruments and sensor systems installed on the seabed and integrated into the volcanic surveillance network of the whole territory [95].

Currently, marine monitoring networks include geophysical and geochemical sensor systems employing conventional hydrophones. Fiber optic sensor technology represents a valid alternative solution to the standard hydrophones also in this application scenario. In particular, the underwater operation poses specific challenges to conventional sensors in terms of maintenance efforts and complexity of the telemetry systems. On the other hand, optical fiber systems allow easily remote operation over long distances with a consequent decrease in complexity [78,96]. Additionally, optical fiber-based sensors are completely “passive”, providing a simpler maintenance.

In the literature, there are several proposals of optical fiber hydrophones with different transduction principle and performances. Indeed, the first idea of using an optical fiber wrapped around a compliant mandrel for developing acoustic hydrophone dates back to 1977 [97]. In the following years, many research groups have proposed different configurations of fiber optic sensors, such as interferometric schemes [98], coated FBG-based sensors [99,100] or fiber lasers to develop hydrophones for underwater applications [101]. Nonetheless, as highlighted by a recent review from Meng et al. [102], there is a clear lack of low frequency hydrophones (specifically operating below 40 Hz), nor are they deployed on the seabed and demonstrated in operative conditions for vulcanological monitoring. To note, we reported in 2022 a field demonstration of fiber optic hydrophones for earthquakes monitoring in real operative scenarios [71]. Several works deserve to be mentioned for the achieved performances, even if most of them have been developed for SONAR applications (i.e., in the “acoustic” frequency range) and not for seismic monitoring applications (i.e., <100 Hz). Kilic et al. demonstrated an interferometric hydrophone based on a Fabry–Perot constituted of a photonic-crystal reflector suspended on a single-mode fiber tip [97]. The hydrophone responsivity was measured from 100 Hz to 100 kHz, demonstrating a sound-pressure-equivalent noise spectral density down to 12 μPa/Hz^1/2^, a flatland wider than 10 kHz and very low distortion. Foster et al. [103] developed a distributed feedback fiber laser (DFB-FL) hydrophone device, with an acoustic responsivity of 107 dB re Hz/Pa over the bandwidth from 100 Hz to 5 kHz. Such hydrophone was deployed at a 33 m deep seabed for a field test at the south coast of Australia and was tested using a calibrated acoustic source in the frequency range from 500 Hz to 3 kHz [104]. In 2014, H. C. Gu fabricated a linear dragging array containing four fiber laser hydrophones which use DFB-FL as sensing element [105]. The underwater acoustic detection experiment and towing test were conducted in the lake of Mogan Mountain, showing a responsivity of −135 dB in the frequency range from 20 Hz to 1000 Hz. Various configurations of interferometric FOH based on idea of Bucaro [106,107,108] have been proposed too. We also designed an FOH to operate in an acoustic frequency range up to 10 kHz and then developed a fiber optic towed array [109]. Lavrov et al. [110,111] carried out experimental trials of interferometric hydrophone using FBG or Faraday rotation mirrors at the lake in Russia. However, even if the numerous works present in the scientific literature confirm the strong potential of fiber optic interferometric hydrophones for underwater seismic monitoring, they do not provide either a characterization at seismic frequencies (not reliable in small tanks or laboratory experiments [112]) or a demonstration of operation in a real scenario on the seabed for earthquake monitoring applications.

### 3.4. Design and Field Demonstration of the Lab on Fiber Seismic

The fiber optic hydrophone (FOH) has a cylindrical shape, and it is constituted of a composite mandrel. In Figure 16a,b, we display the schematization of the FOH. The structure of the hydrophone relies on a plastic shell filled with oil and a compliant solid core. A steel rod at the center serves for supporting the structure. Around the plastic shell, an optical fiber is suitably wound.

An acoustic wave with a wavelength larger than the hydrophone size behaves similar to a hydrostatic pressure for the hydrophone; that is, a uniform force is exerted on the external surface such as to cause a deformation of the complaint mandrel layer. The deformation involves either an expansion or a compression of the mandrel, which results in a strain in the optical fiber around it [71,109].

We used two FBGs at different wavelengths to spectrally mark the beginning and the end of the fiber coil. A Michelson interferometric scheme (MultiZonaSens, Optics11 [113]) is used to retrieve the fractional length changes in the fiber coil versus time. A “dummy” hydrophone, featuring a pressure-insensitive mandrel wound by an optical fiber is used as a reference arm.

The responsivity of the FOH under the applied pressure can be expressed:(1)$$S_{l}=\frac{∆l_{f}}{P}=\frac{l_{f}}{R_{m}}\cdot \frac{∆R_{m}}{P}$$
where ∆*l_f_* is the fractional variation in fiber length with respect to the applied pressure *P*, and $∆R_{m}$ is the radial displacement. To determine $∆R_{m}$, which enables us to evaluate the responsivity, we performed a full 3-D numerical simulations by finite element method [71]. Specifically, the hydrophone design was devoted to obtaining a sensitivity as high to possess self-noise levels below sea-state zero and to offer a robust package for underwater operation. The final sensing configuration features a double layer of fiber coil covering the plastic shell for a height of 30 mm and a mandrel radius of 35 mm. The corresponding optical fiber length is about 126 m. The numerical analysis reveals that such configuration is able to provide a mean radial displacement of 0.1 nm suitable to reach a responsivity of 347 nm/Pa.

A prototype of the FOH sensing system was realized and integrated into the INGV seismological network for monitoring the geodetic activity of the “Campi Flegrei” caldera. The complete sensing system comprises two FOHs, interrogated by one optoelectronic unit. Specifically, two FOHs were fabricated and installed on a seafloor module connected to one of the geodetic buoys present in the open sea in the Gulf of Pozzuoli (see Figure 17a). The two hydrophones are identical and collocated in the submarine module, allowing for redundancy and multiplexing proof. The sensing hydrophones are paired with reference dummy hydrophones (see Figure 17b). The FOHs were anchored to the submarine module (Figure 17c) and then deployed on the seabed. The interrogation system is placed on the buoy. The overall system is schematized in Figure 17d [71]. The deployment of the optical seismological monitoring system was set up and operative in December 2020.

The Campi Flegrei caldera, located northwest of the city of Naples (see Figure 17e), is currently active. Therefore, we have taken advantage from the caldera seismicity in order to characterize the responsivity of the FOH and to assess the capability of the FOH to detect earthquakes. In particular, a seismic swarm featuring several events of low magnitude was recorded from 18 to 21 December 2020. The events are characterized by different amplitudes and spectral ranges, and they have been used as a “natural” source of seismic waves to characterize our FOH sensors by exploiting a commercial PZT hydrophone as a reference. The sensing system exhibited a responsivity of about −300 nm/Pa and an average noise floor level down to 100 µPa/√Hz in the 1–80 Hz frequency range. As a representative example of the FOH response, in Figure 17f we show a 4 s window of waveforms of the same seismic event recorded on the 19 December 2020 at 22:54 UTC, with a magnitude of 2.7 Md, by the FOH and the reference PZT hydrophone. It is noteworthy that the time domain of seismic signals recorded by the FOH and the PZT hydrophone are very similar (i.e., the mutual correlation was approx. 85%).

We have reported the results obtained from the experimental field of innovative fiber optic sensors for land and underwater seismic monitoring. Developed fiber optical sensors are promising alternatives for seismic wave detection and marine environmental monitoring systems. The performances are competitive with traditional sensors. Furthermore, the characteristic of these sensors makes it possible to be positioned at long distances from the interrogation system, while PZT systems require a careful tradeoff between cable length and amplification, limiting the distance between the sensors and the telemetry station.

## 4. Fiber Optic Sensors for High Radiations Environments

A High Energy Physics (HEP) apparatus, such as the particle accelerator complex and particle detectors operating at the European Organization for Nuclear Research (CERN) Large Hadron Collider (LHC), works in very complex environmental conditions requiring a constant monitoring of physical quantities such as temperature, structural deformation, relative humidity, magnetic field and ionizing radiation. Indeed, all these peculiar experimental areas are under constant monitoring with the main scope of having information about of the working condition of all the subsystems. The monitoring became crucial for all the equipment that are temperature dependent and/or must work under particular thermal conditions [114,115]. The need to develop new instrumentation for ambient parameters monitoring, capable of withstanding and operating at high radiation doses, becomes more crucial in view of the High Luminosity Large Hadron Collider (HL-LHC) project [116,117] which aims to an upgrade of the most powerful accelerator of the world. The novel configuration, indeed, relies on several key innovative technologies increasing the accelerator luminosity by a factor of five beyond its design value, thus resulting in a ten times higher level of radiation, in terms of both ionizing dose and particle fluence.

In the next sections, we provide an overview of the research activities carried out by our multidisciplinary group, currently running in collaboration with CERN, concerning the development and the application of fiber optic sensors (FOSs) in HEP environments.

### 4.1. FOS for Relative Humidity Monitoring in HEP Environments

RH monitoring has a significant impact in various application fields, and many sensing solutions have been proposed according to the specific applications [118,119]. Electronic humidity sensors still cover the main part of the sensors market as their fabrication technology is well established. With the advent of optical fiber technology, a considerable effort of the research has been focused on the development of FOSs for humidity sensing due to their advantage in terms of reduced size and weight, multi-parametric sensing, easy cabling, electromagnetic immunity and radiation tolerance required for example in cases of application in harsh environments. A wide range of approaches of FOSs has been proposed in the literature over the years, as recently reported in [120]. In 2011, our multidisciplinary research group has been involved in the development of FOSs for RH monitoring in the Compact Muon Solenoid (CMS) experiment, one of the four large detectors of the CERN accelerator. Due to the high radiation level resulting from the accelerator operation, the CMS tracker sensors operate in an experimental environment with sub-zero temperatures. Under these conditions, monitoring the ambient parameters, notably temperature and humidity, is vital in order to avoid vapor condensation, which would damage both the sensors and the complex and expensive readout electronics. Any humidity sensor to be introduced in the CMS detector volume should comply with the requirements in terms of radiation resistance up to 1 MGy, insensitivity to magnetic field, correct operation at temperature down to −20 °C and at low humidity values and a reduced number of wires for operation [121]. Several miniaturized commercial sensors have been exploited in the CMS experiment, but none of them really satisfy the above-mentioned full list of requirements. Indeed, the performance of any commercial RH sensor operating in HEP detectors has been demonstrated to be degraded in time due to high radiation doses [122]. In this scenario, FOSs appear to be a good alternative to the conventional instruments.

In 2013, Berruti et al. proposed for the first time a Fiber Bragg Gratings (FBGs)-based hygrometer coated with a micrometer-thick polyimide (PI) overlay for RH monitoring in CMS experiment [56]. The hygroscopic material, which covers the grating, swells upon water molecule adsorption, thus inducing mechanical strain on the grating itself, which results in a Bragg wavelength shift [54,55]. Due to the intrinsic FBG sensitivity also to temperature (T) variations, the final proposed solution consisted of a thermo-hygrometer made of two coupled FBGs, one PI-coated and one bare, for RH and T readings, respectively, as shown in Figure 18a. Experimental tests have been performed in the range [0–60]% RH at different temperatures, and the response of the FBG sensors toward humidity was found to be linear also at temperatures below 0 °C (Figure 18b) [56,123,124].

Incremental γ- irradiation campaigns up to 210 kGy have been performed to investigate the FBG thermo-hygrometers radiation tolerance characteristic. As shown in Figure 18c,d, it was found that the proposed sensors survive after the exposure to strong γ-ionizing radiation doses, and their sensing performance is not affected by radiation. A pre-irradiation step at doses higher than 150 kGy, before the installation in high radiation environment, is needed to greatly reduce their sensitivity to further irradiation [123,124].

Based on the very promising results, 72 FBG-based thermo-hygrometers, organized in multi-sensors arrays, have been installed in CMS since 2013 in critical areas of the experiment where power, read-out and cooling services are distributed to the Tracker detector [125], as it will be shown in the Section 4.2.1.

Despite the correct operation in the CMS experiment confirming the strong potential for the application of RH monitoring in real harsh environments, such technology was found to be not immune from some limitations, mostly related to the cross-sensitivity of coated FBG to T and RH and to the chemical instability of the polyimide coating [123,125]. For this reason, in 2014 our investigations moved towards the development of a second generation of RH sensors based on Long Period Grating (LPG) technology [123,126]. The final proposal consisted of a high-sensitivity LPG sensor coated with a finely tuned titanium dioxide (TiO_2_) thin layer (~100 nm thick). The processes of water absorption/desorption in the hygrosensitive coating in the presence of RH variations modify its refractive index, thus creating a spectral variation and amplitude change in the LPG attenuation bands, as results from Figure 19a.

RH characterizations in the range [0–75]% RH at different temperatures were carried out to assess the sensors performance in real operative conditions. Experimental results (Figure 19b,c) demonstrated the very high RH sensitivities of the proposed device (≈1.4 nm/%RH at low RH), one to three orders of magnitude higher than those exhibited by the FBG-based hygrometers [123].

The radiation tolerance capability of LPG-based hygrometers was also investigated [126]. Interestingly, the TiO_2_-coated LPG sensor was found to not lose its excellent capability to respond to RH changes, and, apart from the radiation-induced dip wavelength shift which must be taken into account, the RH characteristic curve still exhibits the same behavior shown before irradiation (see Figure 20).

Considering such very promising results, in 2020 a prototype system based on TiO_2_-coated LPG sensors was installed in the Inner Detector of “A Toroidal LHC ApparatuS” (ATLAS) experiment during the second long shutdown of the CERN accelerator [114]. Such installation confirmed the interest of the HEP community toward this new technology for RH monitoring in current and future collider detectors.

### 4.2. FOS in Operation in CERN Experiments

For the subsequent discussion, we focus our attention on the CMS experiment [114], one of the four big experiments active on the LHC ring. The monitoring system originally installed in CMS is made of traditional electronic sensors [114]. The already significant number of detectors and electrical wiring installed at the CMS do not facilitate the installation of additional monitoring systems. Moreover, during the operation of LHC, the high level of radiation and the magnetic field are often not compatible with a good operation of conventional electronic sensing devices. The FBG technology, unknown to the CMS community at the time of the design of the detector, represents a valuable and innovative answer to the request for additional monitoring systems to be inserted in the complex structures of the CMS detector where cabling easiness plays a key role. Indeed, the FBG technology allows the realization of wide monitoring quasi-distributed systems with an elevated number of sensing points using a single optical fiber. Monitoring systems based on the FBGs technology have been installed in the underground site of the CERN CMS experiment since 2009 [127]. The whole CMS FBG monitoring system has been expanded during the past years and, in the present configuration, is composed by nearly one thousand FBG sensors, covering the CMS experiment from the outer to the innermost part. The readout system of these FBGs is based on the use of several interrogators whose output is perfectly integrated in the CMS Detector Control System [128]. The scheme of the data management for the system is depicted in Figure 21.

#### 4.2.1. Examples of FBG Monitoring System within the FOS4CMS Project

##### HF Raisers and CASTOR Platform Structural Monitoring

The 3.8T CMS magnet cycles induce strain deformation on the detector structure. A strain monitoring system composed of eleven FBG sensors has been installed on the HF riser structure. Ten FBGs have been glued on the structure to measure the strain deformation, while one has been left free to be used as thermal compensator [127]. The strain sensor layout is shown in Figure 22. During a B-field rump up, the IP-side of the structure is subjected to a compression while the non-IP side is subjected to a decompression. The CASTOR detector is located on a platform anchored to the HF raiser structure. The mechanical deformation acting on these structures must be monitored to guarantee the safe operation of the detector. To this end, an FBG strain monitoring system has been installed on the CASTOR platform structure. The layout of this monitoring system is depicted in Figure 23. Therefore, at first, a calculation has been performed with zero magnetic field for the gravity-only distortion that was validated by the FBG data.

From the data collected with these FBG monitoring systems, the morphing of the entire structure, composed by HF raisers plus CASTOR platform, was performed and is shown in Figure 24. The calculations showed that despite the relatively large shear displacements (~1.5 mm), the rotation center coincides with the internal beam pipe support; therefore, the beam pipe is not subjected to destructive forces during the magnet ramps.

##### Cavern Temperature Monitoring

In 2011, during LHC Run1 period, a temperature monitoring system for the whole CMS Experimental Underground Cavern (UXC) was installed [127]. It is composed of a two-kilometer-long SMF-28 fiber with 60 FBG sensors, whose layout is shown in Figure 25. The system has been running 24/7 since the installation, and the data are displayed in the CMS control room allowing the monitoring of the thermal gradient of the whole CMS experimental cavern.

##### RPC Endcap Temperature Monitoring

The temperature is an essential parameter to be monitored for the correct understanding of the performances of the RPC detectors. Hence, an FBG based temperature monitoring system was designed to have reliable and robust measurements of the RPC detectors’ temperature during the CMS data taking. On each chamber on the disks RE ± 2, RE ± 3 and RE ± 4 we have placed FBG temperature sensors, that is to say 72 sensors per disk on a single fiber, for a total of 432 FBG sensors. The layout of the system for one disk is shown in Figure 26. These sensors were installed during LS1, and the system is on data taking 24/7, and data are available in the RPC detector control station in the CMS control room.

##### Tracker RH Monitoring

After a dedicated R&D phase, well detailed in the previous section, a full network of 72 optic fiber-based sensors, each one formed by an FBG temperature sensor and an FBG relative humidity sensor, has been installed from the end of 2013 in the critical areas of the CMS Tracker end-flange for constant distributed thermo-hygrometric monitoring [128]. During the LS1, the FBG thermos-hygrometer has been compared with a network of conventional sensors allowing for validation of the FBG measurements in terms of both temperature and relative humidity, in a region where the concentration of power cables and cooling pipes creates strong local gradients. The full network of FBG-based thermo-hygrometers installed in the experiment has been continuously working from the restart of the LHC operation for the Run2, providing distributed monitoring of the environmental conditions in the Tracker volume, preventing the risk of local condensation and ice formation. An example of temperature, relative humidity and dew point reconstruction is shown in Figure 27.

#### 4.2.2. Central Beam Pipe Monitoring System

The CMS central beam pipe is a key element of both the LHC ring and the CMS experiment, since it is the place where the proton–proton collisions occur during the LHC operation. The pipe is made of a Beryllium tube section, 3 m long with a central diameter of 45 mm and only 0.8 mm thickness wall, sealed on the two extremities with two conical aluminum (Al EN AW-2219) sections, each 1.5 m long with an external diameter of 65 mm. It must stand to an extreme vacuum condition (down to 10 mbar), and, at the same time, it must not interfere with the particle resulting from collisions. The CMS tube needs to be continuously monitored to obtain information about its structural and health state. It is mandatory that any monitoring system to be installed on the BP must not interfere with the particle detectors wrapped around the beam pipe. Radiation immunity represents one of the most important specifications required to a monitoring system operating in HEP environment, while other needs are low complexity layout, multiplexing and multi-parameters measurement capabilities. On the basis of these technical specifications, a monitoring system based on the FBG technology, that has been called i-pipe (improved beam pipe), was designed and installed on the central BP of the CMS experiment to monitor its thermal conditions and unpredictable mechanical deformations [128,129]. The i-pipe system consists of four naked glass SMF-28 fibers placed longitudinally along the cardinal positions on the BP. Each fiber is an array of 16 FBG sensors: seven are glued on the BP to measure the local strain, and the remaining nine are left unglued but in contact with BP in order to work as local thermometers and as temperature compensators for the adjacent strain sensors. A schematic representation of the sensor distribution along the BP is depicted in Figure 28. The i-pipe system has proved to be an important innovation in the framework of the Structural health monitoring of crucial parts of the CMS detector and LHC beam pipe [129].

The i-pipe system data analysis [130] demonstrated the complete fulfillment of the initial technical specs in terms of monitoring and quantifying any deformation induced on the CMS central beam pipe during the experiment and LHC operations. In Figure 29, the thermal profile during varius CMS opration status, as recorded during LHC Run2, is reported.

**Figure 28 sensors-23-03187-f028:**
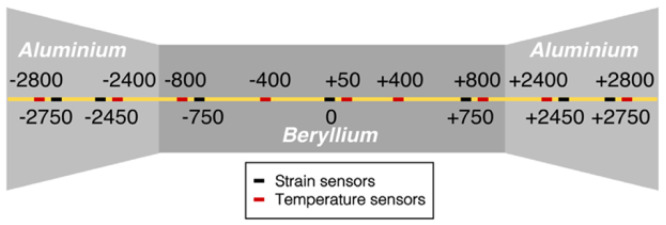
Longitudinal layout of the fiber optic monitoring system installed on the CMS central BP. The quotes indicate the positions, relative to the center of the central BP of the FBG sensors. All the quotes refer to the FBG sensors of a single fiber, with respect to the center of the central BP. Four fibers have been installed in the cardinal position of the BP (adapted from Ref. [131]).

### 4.3. FOS for Dose Monitoring

In recent years, a great effort has been made to demonstrate the applicability of FOS for ultra-high dose levels monitoring [132,133]. The solution currently employed in LHC for dose monitoring is represented by Radiation Monitoring dosimeters (RadMon) [134], which has been demonstrated to be not capable to work in the future HL-LHC radiation environments. For this reason, the investigations of the authors have been addressed to the development of innovative optical fiber-based dosimeters. Optical fiber dosimeters can be divided into intrinsic and extrinsic categories. In the first configuration, including FBG and LPG-based solutions, the optical fiber acts both as sensing and light-guiding component [135]. The study of the radiation effects on optical fiber properties, in terms of creation of point defects in silica and displacement damage, has been the core topic of many publications [136,137]. Certainly, massive knowledge on this topic has been collected over the years, but still not all the factors involved are completely clear and known. A complete review about the recent advances in radiation-hardened optical fiber-based systems was published in 2018 by Girard et al. [138], highlighting the potential and the future challenges of this technology for harsh environments applications. It is well known that radiation alters the fiber properties by creating point defects in silica-based material, due to ionization or displacement damage processes leading to structural modifications in the pure or doped silica matrix of both fiber core and cladding [137,139,140]. Point defects are responsible of the two main radiation-induced macroscopic effects such as radiation-induced attenuation and radiation-induced refractive index change, which degrade the properties of optical fibers when exposed to radiation. Nevertheless, results reported in the literature demonstrate that making appropriate choices related to the fiber composition, some devices have notable radiation resistance [136], suggesting their use for sensing applications in high radiation environments, while others demonstrate vulnerability to radiations [139], thus recommending them for radiation dosimetry.

In extrinsic configurations, the optical fiber is a guide for the light traveling to and from a radiation sensitive material. The mostly exploited principles for extrinsic dosimeters are based on scintillating materials [141], thermo-luminescence [142] and optically stimulated luminescence [143]. It is worth noting that the use of poly(methyl methacrylate) (PMMA) as sensing element provided an important contribution to the development of fiber optic-based dosimeters [132,133,135,144].

#### 4.3.1. LPG for Dose Monitoring in HEP Environments

##### LPG under Gamma and Neutron Irradiation

The effects of high doses of gamma radiation on standard and specialty optical fibers have been investigated by the authors in different studies using long period gratings [145,146,147,148]. Specifically, few of the fibers considered are the following: standard single-mode Ge-doped Corning SMF28; radiation resistant Nufern R1310; specialty Fiber-A (confidential manufacturer) designed for high pressure, high temperature and corrosive surroundings; and radiation hardened Fiber-B (confidential manufacturer) with a pure-silica core and F-doped cladding. The fabrication of LPG in most of these fibers is not trivial due to the lack of photosensitivity, and the electric arc discharge method was selected due to the high degree of flexibility for the mentioned aim. The period of the LPGs was designed in the range of 650 ± 25 µm to have coupling with the same order cladding modes in the near infrared wavelength range, around 1560 nm [149,150,151]. The gamma irradiations were conducted using a Co^60^ source stored in a pool at an industrial irradiator of the “Horia Hulubei” physics institute (Magurele, Romania). The irradiation conditions were the following: room temperature, 0.2 kGy/h dose rate, time range of 150–300 h and a total absorbed dose within 25–35 kGy. The experimental analysis focused on the real-time measurement of the LPG resonance wavelength shift and optical transmission of the fiber, as a function of the irradiation time/dose. The results are comparatively illustrated in Figure 30a,b for wavelength shift and power changes, respectively, whereas the irradiation profiles are detailed in Figure 30c.

The main outcome is that a wavelength shift towards higher wavelengths can be observed for all the fibers with different magnitudes. Higher sensitivity was obtained in the first hours, i.e., for doses up to 10 kGy, and a saturation behavior occurred after 15 kGy. Specifically, wavelength shifts of 3.7 nm and 6.7 nm were measured for the LPGs in SMF28 and Nufern R1310 fiber, respectively, after a total dose of 35 kGy. A significant wavelength shift of 5.7 nm was also found for Fiber-A LPG after a 27 kGy dose, whereas an almost negligible change of 0.2 nm was only experienced by Fiber-B LPG after 30 kGy. As far as the changes in transmitted optical power (in range 1510–1520 nm) are concerned, few dB variations increasing with dose were observed in the case of SMF28 and Fiber-A samples, while negligible changes were experienced by Nufern R1310 and Fiber-B as a result of their radiation resistant/hardened designs. Such experimental campaign was further enriched with numerical simulations providing an estimation of the radiation induced refractive index changes, being the main responsibility of the LPG wavelength shifts [152], such values were found to be in the order of 1.5–2.5 × 10^−5^ for the radiation sensitive fibers and one order of magnitude less for the pure-silica core fiber. Finally, the radiation-induced changes on the grating temperature sensitivity were also evaluated.

Very recently, the impact of gamma irradiation on the optical properties of a wide selection of commercially available optical fibers was conducted in [153]. The single mode optical fibers under consideration were doped with the following elements: Ge, B, F and P. The fibers with LPGs were subjected to a gamma dose rate of 2.6 kGy/h up to a total dose of 52 kGy. A reflective configuration of the gratings was here for the first time in such domain. Moreover, as a novelty, the recovery effects were monitored in real-time for the following two days. The wavelength shift and power change were observed both during irradiation and recovery periods. The results are reported in Figure 31, in terms of wavelength shift as a function of the time. The grating response changed significantly with the fiber model. The highest radiation sensitivity was experienced by B/Ge fiber with a 10 nm shift; moreover, the least recovery effect was observed for the same fiber. However, concerning the power changes, a strong irradiation-induced response was experienced by P-doped fiber with negligible recovery. A similar set of LPGs was also the subject of another important investigation, which was conducted for the first time in [154], where the gratings were subjected to a mixed neutron-gamma field in a TRIGA type nuclear reactor at the Nuclear Research Institute ICN in Mioveni, Romania. The fibers used for LPG inscription were the same of those in [147], except that in this case the radiation hardened pure-silica core fiber was a Prysmian DrakaSRH. The irradiation conditions were the following: time 2 h, gamma dose rate of 9 Gy/s and gamma total dose of about 65 kGy, mean neutron flux of 1.25 × 10^12^ n/(cm^2^∙s) and a 9.18 × 10^15^ n/cm^2^ neutron fluence.

Wavelength shifts of the LPGs were measured in real-time during the exposure and are reported in Figure 32a, while the irradiation profiles can be observed in Figure 32b. Here, the wavelengths increase with the gamma dose and neutron fluence with higher sensitivity during the first minutes and reach saturation after about half hour, i.e., corresponding to a gamma dose of about 15 kGy and a neutron fluence of 2 × 10^15^ n/cm^2^. At the end of the irradiation, the following situation was thus obtained: the shift for LPGs in SMF28, Fiber-A, Nufern and Draka were equal to, respectively, 6.4 nm, 9.0 nm, 11.8 nm and −0.4 nm. It is worth highlighting that these values are in good agreement with the trends measured during only gamma irradiation of the same samples performed in [147]. The estimation of the radiation induced refractive index changes within 2.5–4 × 10^−5^ and the evaluation of temperature sensitivity post irradiation were also subject of the study.

##### LPG under Proton Irradiation

In the frame of the project concerning the development of optical fiber-based RH sensors in the CERN experiments, the authors investigated the response of LPG-based devices under strong doses of radiations. Taking into consideration the radiation field at CERN HL-LHC, characterized by the presence of both leptons and hadrons of different kind, masses and energies, protons seemed to be better candidates to simulate the combined effect of Total Ionizing Dose (TID) and Displacement Damage (DD), typical of this field.

In 2020, Berruti et al. presented a complete study of uncoated LPGs written in photosensitive single-mode B-Ge co-doped optical fiber using a point-to-point technique, exposed to high fluence of protons (4.4 × 10^15^ p/cm^2^) up to a total absorbed dose of 1.16 MGy [155,156]. By combining the experimental results with numerical simulations, the variations of the major parameters affecting the LPG response during the ultra-high dose proton exposure have been evaluated. It was demonstrated that the irradiation exposure induces a variation in the core effective refractive index, which is responsible for a resonance wavelength red shift (Figure 33a). At the same time, a relevant decrease in the refractive index modulation pertaining to the grating was estimated, leading to a reduction of the resonant dip visibility ((Figure 33b).

Collected results paved the way for the development of a novel class of miniaturized optical dosimeters potentially providing a single device with high sensitivity from very low to ultra-high ionizing doses and opening new perspectives for promising multi-parametric sensing applications of the LPG technology in high radiation environments.

#### 4.3.2. LOF for Dose Monitoring in HEP Environments

Based on a previous demonstration of an LOF device carried out by our multidisciplinary research group [157], where the key idea was to transform a ‘simple’ OF into a miniaturized and multifunctional platform for specific applications through the integration of functionalized materials and components defined at micro- or nanoscale [86,87,158], Quero et al. presented in Ref. [159] an LOF dosimeter for ultra-high dose monitoring in HEP environments based on a metallo-dielectric resonator realized on the optical fiber tip. The proposed LOF nanostructure consisted in the superimposition of two metallic gratings separated by a radiation sensitive PMMA dielectric layer (Figure 34).

To test the response of the sensor to ultra-high levels of radiation, the first irradiation experiments have been performed at the CERN high-energy proton irradiation facility (IRRAD) with 23 GeV protons for a total accumulated dose of 1.8 MGy in 1 cm^2^ of silicon. As shown in Figure 35a, the protons exposure at high doses did not produce any spectral strong degradation, thus demonstrating the LOF technology radiation resistance feature. Differently, Figure 35b shows a clear LOF resonance migration towards lower wavelengths (of about 1.4 nm) with increasing dose level. At the same time, a slight increase in the power baseline of about 0.16 dBm was observed. As supported by the literature [160,161] and our numerical simulations, a reduction of the PMMA thickness is consistent with the observed spectral trends. Preliminary experimental results demonstrate the effectiveness of the proposed LOF device as potential dosimeter at MGy dose levels. In particular, a maximum dose of 1.8 MGy has been measured with a sensitivity of −0.6 nm/MGy and a resolution of 16.7 kGy.

#### 4.3.3. Optical Fiber Technology and Radiochromic Films for Dose Monitoring

The potential deriving from the integration of fiber optic technology with materials responsive to ionizing radiation has also been explored using radiochromic films (RCFs). RCFs are widely employed as dosimeters thanks to their ability to modify the structural characteristics of their crystalline sensitive element when exposed to ionizing radiation [160]. This microscopic phenomenon is reflected in a color change (darkening) of the film that can be related to the absorbed dose [161,162,163,164]. Several types of films are currently available on the market, covering a very wide range of dose from mGy up to hundreds of kGy [165]. RCFs are considered as a reliable technique for accurate dose assessment and quality checks in a variety of applications of radiation physics, ranging from medical physics to beam diagnostics, studies of radiation damage of electronic devices, radiation processing and radiation-induced sterilization [162,166,167,168]. Typically, the dose measurement by RCF implies the evaluation of the blackening level of the film, compared to the same film before exposure to radiation. The measurement is performed using densitometers, flatbed scanners and more rarely spectrophotometers [169,170,171]. Unfortunately, all these methods do not allow measurements of the trend of dose in time but only integrated dose, since the color change in the film is read only after the exposure to radiation, thus representing a major limitation in many applications. Although there are dosimeters capable of measuring the dose in real time (e.g., gas-filled chambers, scintillation detectors, phosphor screens and semiconductor detectors), the online reading of an RCF would represent a significant step forward in application scenarios, possibly in combination with one of the aforementioned techniques. At present, there have been very few attempts to read the dose in real time by RCF [172,173]. In this framework, in collaboration with the Istituto Nazionale di Fisica Nucleare (Sezione di Napoli) and CERN, we have developed an innovative optical fiber-based RCF reading method that has the merit of providing a real-time measurement of the dose with very low uncertainty [174,175,176]. Unlike the previously reported approaches, in which the material sensitive to the dose is deposited onto the tip of the fiber providing a single-use probe, we adopted an easier and disposable technique in which the RCF is simply fixed onto the top of a fiber optic probe and finally replaced with a new piece of film after the irradiation.

Moreover, the same probe can serve for very different dose ranges by merely inserting the more appropriate RCF model. The typical schematic of the optical fiber based RCF reading method is shown in Figure 36. The film is once crossed by the incoming light (provided by a broadband light source), which is backscattered by a reflective material and, after passing again through the film, is finally collected by a spectrophotometer. Both the illumination of the film and the collection of the backscattered light crossing the film are ensured by two optical fiber bundles, which are connected to the light source and the spectrophotometer, respectively. This configuration can be easily modified to allow the reading of the dose absorbed by the RCF if the reflecting material is made in such a way as to be transparent to the dose, or it can be completely eliminated if the RCF itself is equipped with a reflective side [174]. During the exposure to ionizing radiation, the optical changes in the film are detected in real-time by the collecting fiber and sent back to the spectrophotometer for data analysis. The method was tested for several RCF types, showing excellent performances in terms of accuracy and precision. The results of the irradiations of two types of films (EBT3 and XR-QA2 Gafchromic models) to ^60^Co-γ rays at two dose-rates (2.59 and 0.10 Gy/min) are reported in Figure 37. The increase in absorbed dose produces a decrease in amount of light collected by the spectrophotometer (Figure 37a,c), due to the radiation induced darkening of the film. Each RCF model was tested in its own typical dose range, and the spectra were analyzed at different wavelengths depending on the range of interest (Figure 37b,d).

The obtained calibration curves were finally validated by exposing additional films (from the same batch) to known doses and comparing the response to the nominal doses, allowing the evaluation of the corresponding uncertainties and showing a maximum uncertainty of 8% from 30 mGy, 4% from 100 mGy and 2% from 2.5 Gy on, respectively. The versatility of the proposed method provided by the conjugated use of a broadband light source and a spectrophotometer affords the capability of performing the dose calibration of an RCF by exploring different criteria of data analysis. Based on this concept, Vaiano et al. demonstrated the capability of extending the sensitivity of the EBT3 film (typically saturating over a few tens of Gy with typical reading techniques) to higher dose levels reaching 100 Gy while ensuring low dose uncertainty [177]. In particular, by combining a wavelength-based approach with the monitoring of two characteristic peaks of the EBT3 Optical Density spectrum, the authors were able to measure the dose absorbed by EBT3 films exposed to 1 MeV electron beam and 250 kV X-rays in the range 0.5–100 Gy (Figure 38) with an uncertainty below 4% for doses lower than 5.52 Gy and below 2% for higher dose levels. These results are promising in view of a potential application of this technique in the field of clinical dosimetry at high dose levels. Moreover, the size of the film required for dose measurement is significantly reduced compared to that needed by flatbed scanners due to the use of optical fibers as a reading tool.

## 5. Conclusions and Future Trends

This paper has reviewed our main applications of optical fiber sensors for environmental monitoring. After reporting on some configurations useful for high precision agriculture, we have explored the problems connected with both soil water content and landslide early warning. Then, we have concentrated on a new generation of seismic and vibrational sensors useful in both terrestrial and underwater contexts. At the end, we have discussed some optical fiber sensors for use in radiation environments. The variety of our applications is completed by those reported in refs. [48,49] published in this issue.

In the reported test cases, the proposed solutions are always based on photonic technologies but implemented according in different schemes (point-based grating sensors, distributed sensors, ferrule-top based, etc.). This comes from the fact that the application fields are very diverse, and therefore there is no unique, optimal solution for all of them. A common factor of these solutions is the use of optical sensor technologies combined with proper functional or coating materials, making these sensors suitable for the intended application requirements.

Each proposed solution has specific strengths and weaknesses. For example, the FBG- and LPG-based solutions offer a compact and efficient solution but are subject to cross-sensitivity with few environmental conditions. On the other hand, distributed sensors offer a wealth of information, but they are subject to failure if the fiber breaks at some point.

The results presented in this paper, as well as those presented in refs. [48,49], have driven the creation of some spin-off companies (Often Medica, Optosmart, Optosensing, Biotag) working on the design and application of point-based and distributed optical fiber sensors for industrial and medical applications and the construction of the Center of Nanophotonics and Optoelectronics for Human Health and Industrial Applications (CNOS), which will be completed by April 2023.

The main goals of our future activity can be divided along three main lines:To translate the innovative research results into other market products thus creating new startup companies.To explore new applications to improve both the safety and the security in other fields such as agrifood, antiterrorism, biomedical and precision medicine, the environment and energy saving.To improve the performance of our devices by increasing the use of both the nanotechnology and of the nano materials.

## Figures and Tables

**Figure 1 sensors-23-03187-f001:**
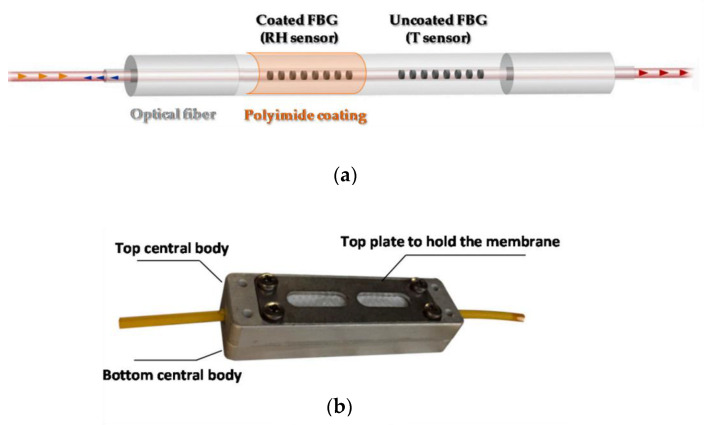
(**a**) Schematic illustration of the proposed fiber optic thermo-hygrometer based on FBG technology. (**b**) Picture of the package developed for the FBG-based thermo-hygrometer (from Ref. [53]).

**Figure 2 sensors-23-03187-f002:**
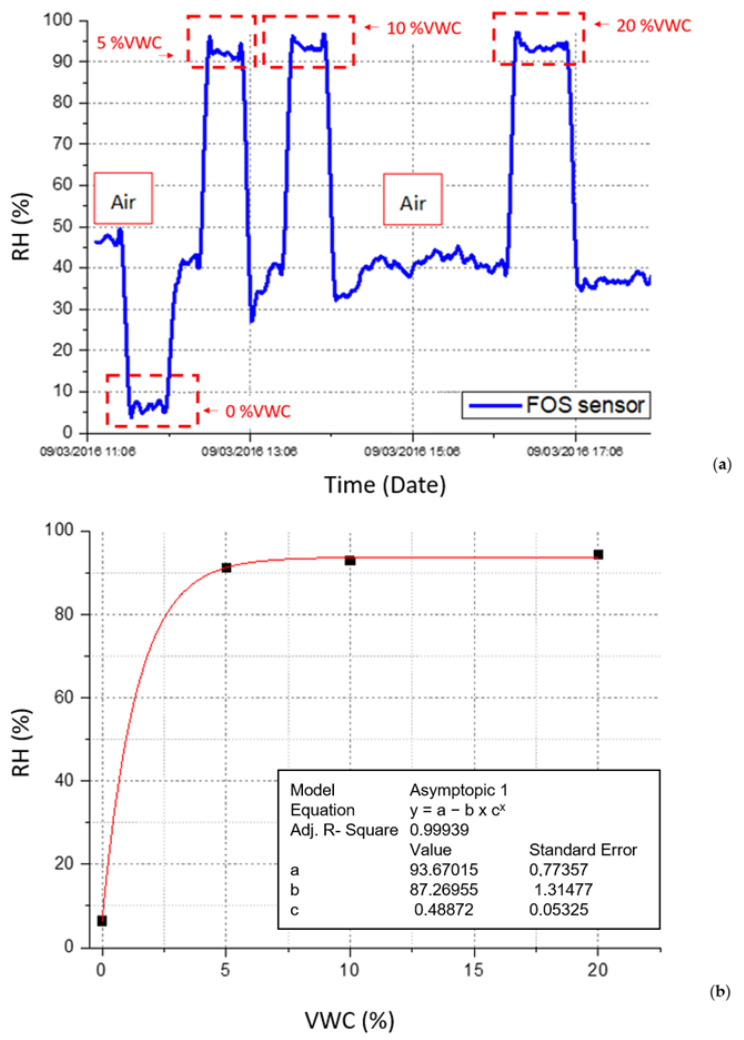
(**a**) FBG thermo-hygrometer response to incremental VWC values; (**b**) RH versus VWC calibration curve (from Ref. [53]).

**Figure 3 sensors-23-03187-f003:**
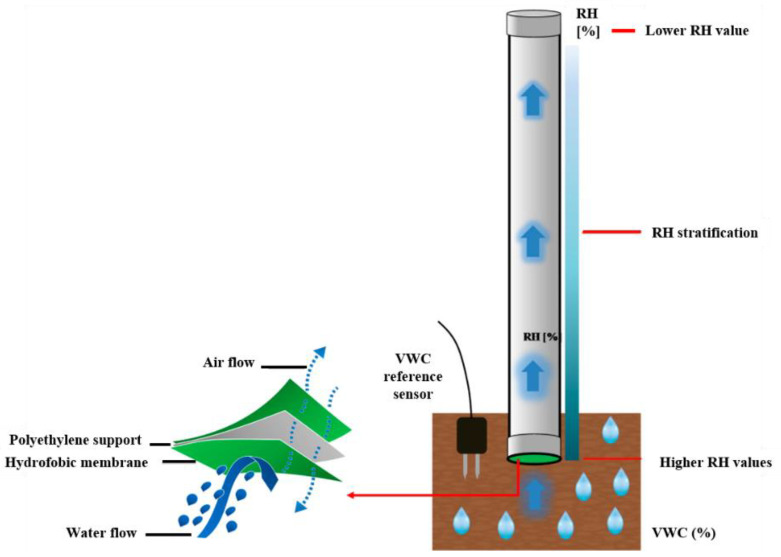
Operation principle of the optimized version of the fiber-optic sensor for soil moisture monitoring (from Ref. [53]).

**Figure 4 sensors-23-03187-f004:**
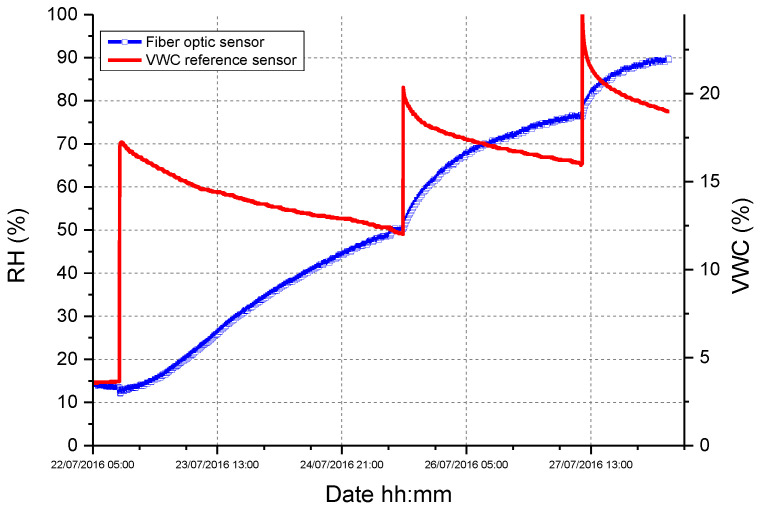
Response of the optimized soil moisture sensor to incremental irrigation steps (from Ref. [53]).

**Figure 5 sensors-23-03187-f005:**
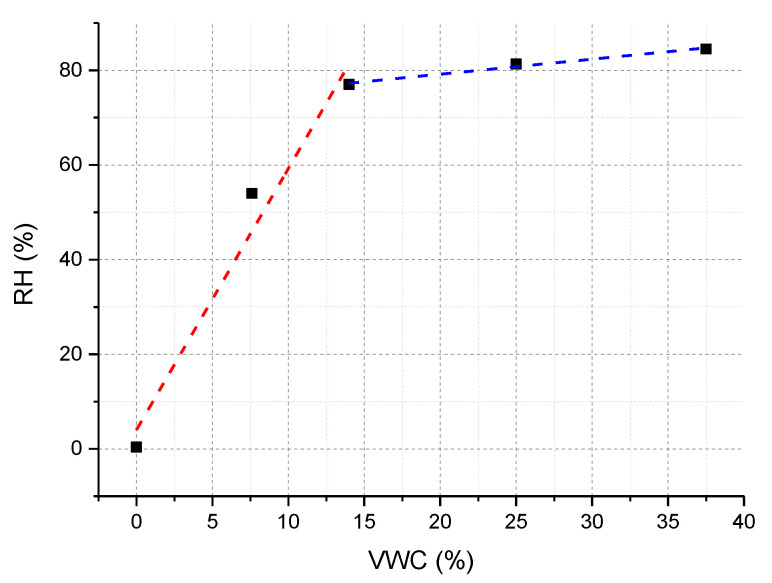
Optimized soil moisture sensor RH versus VWC calibration curve. Linear fitting of the data is reported with red and blue line for low and high VWC range, respectively (from Ref. [53]).

**Figure 6 sensors-23-03187-f006:**
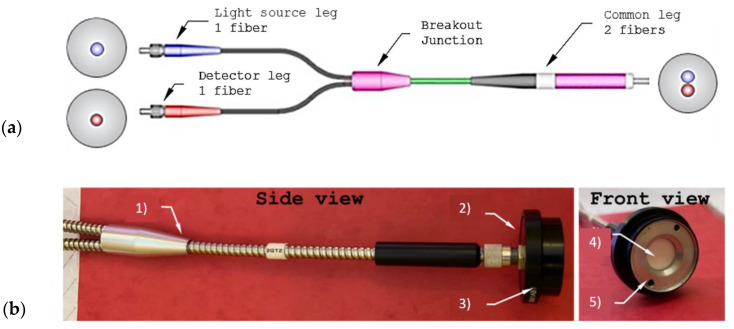
(**a**) Concept design of the proposed bifurcated fibers. (**b**) Realized prototype: (1) optical fiber cable Y-shaped, (2) fiber adapter plate, (3) optic mount, (4) ceramic disk, (5) stainless steel ring. Reprinted with permission from Elsevier from Ref. [59] (the figure is released under a Copyright Clearance Center’s RightsLink^®^ service, Copyright © 2022).

**Figure 7 sensors-23-03187-f007:**
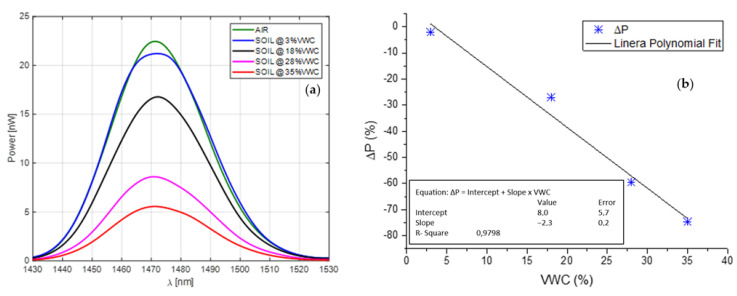
(**a**) Reflected spectra; (**b**) Characteristic curve (ΔP vs. VWC) where ΔP is defined as the reflected power variation at 1470 nm normalized with respect to the initial maximum power. Reprinted with permission from Elsevier from Ref. [59] (the figure is released under a Copyright Clearance Center’s RightsLink^®^ service, Copyright © 2022).

**Figure 8 sensors-23-03187-f008:**
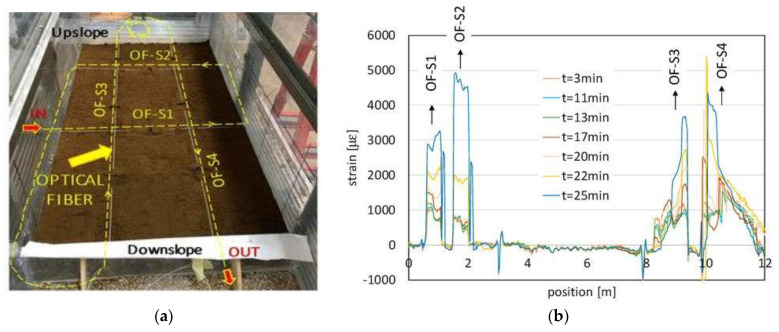
(**a**) Fiber installation during the reconstitution of the slope; (**b**) Strain trends measured by the optical fiber sensor (from Ref. [65]).

**Figure 9 sensors-23-03187-f009:**
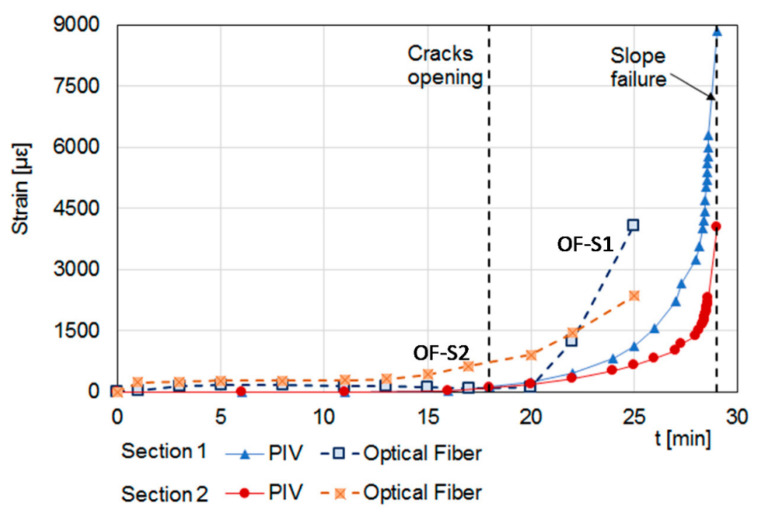
Distributed strain profiles measured by the transverse optical fiber strands and calculated using the digital camera at position of OF-S1 (45 cm away from the downslope) and OF-S2 (30 cm away from the upslope) (from Ref. [65]).

**Figure 10 sensors-23-03187-f010:**
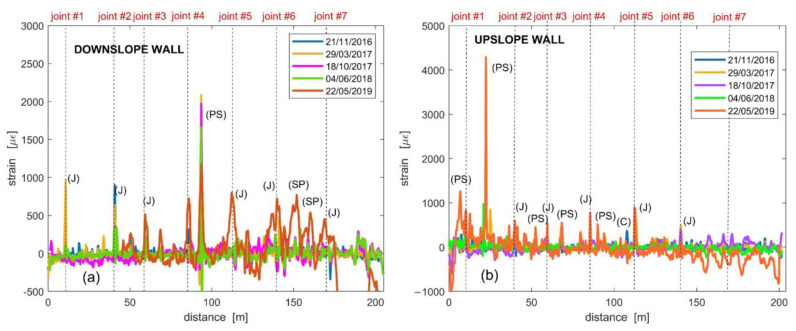
Results of the optical fiber strain measurement along the downslope (**a**) and upslope (**b**) tunnel wall, with indication of the position of joints. J—Joint, PS—local Parget Swelling, SP—Salt Precipitate accumulation, C—Crack (from Ref. [66]).

**Figure 11 sensors-23-03187-f011:**
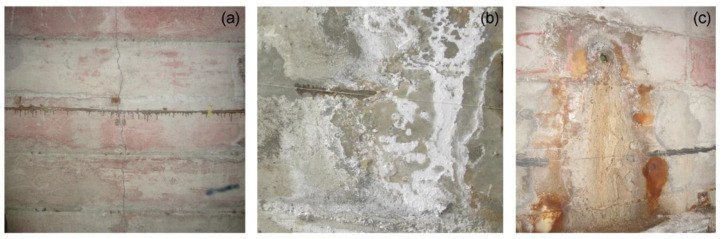
(**a**) Crack on the tunnel wall; (**b**) parget swelling; (**c**) salt precipitate accumulation (from Ref. [66]).

**Figure 12 sensors-23-03187-f012:**
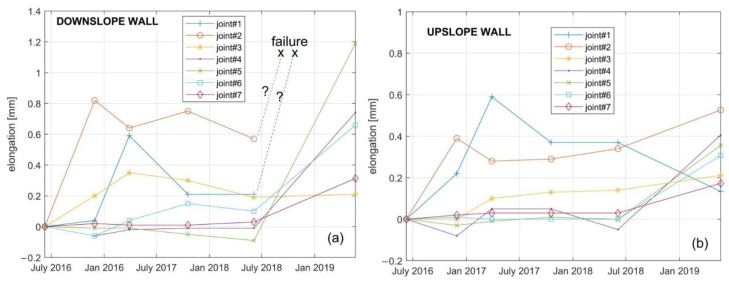
Elongation of the optical fiber computed based on the strains measured across the tunnel joints along the (**a**) downslope or (**b**) upslope sidewall. The symbol “?” indicates that measurements could not be taken due to fiber failure, while the symbol “x” denotes fiber failure (from Ref. [66]).

**Figure 13 sensors-23-03187-f013:**
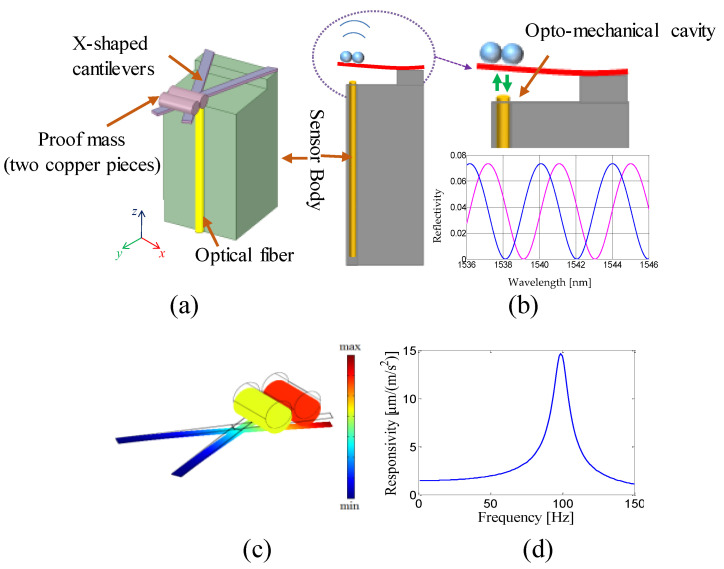
(**a**) Schematic and (**b**) principle of operation of the Lab on fiber seismic sensor. Inset: Schematic picture of the cantilever deflection due to a ground vibration. The Fabry-Pérot air cavity changes due to the effect of cantilever deflection and an interferometer phase shift occurs in the reflectance spectrum. Exemplificative reflection spectrum of the seismic sensor before (blue line) and after deflection (magenta line). (**c**) “Deformed shape” representation of the total displacement when the structure is subject to a vertical acceleration at 5 Hz. (**d**) amplitude of the numerical responsivity of the OF sensor. Reproduced with permission from ref [70] (the figure is released under a CC-BY-SA license http://creativecommons.org/licenses/by/4.0/ (accessed on 12 March 2023)).

**Figure 14 sensors-23-03187-f014:**
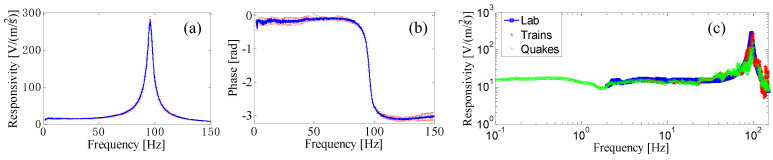
(**a**) Mean responsivity based on ten measurements and relative error bars for the optical fiber sensor in terms of the amplitude and (**b**) phase. (**c**) Comparison among the experimental responsivities of the LOF sensor retrieved by using Lab experiments (blue line) and by using natural sources of vibration like trains (red line) and earthquakes (green line). Reproduced with permission from ref [70] (the figure is released under a CC-BY-SA license http://creativecommons.org/licenses/by/4.0/ (https://www.nature.com/articles/s41598-018-25082-8 (accessed on 27 April 2018)).

**Figure 15 sensors-23-03187-f015:**
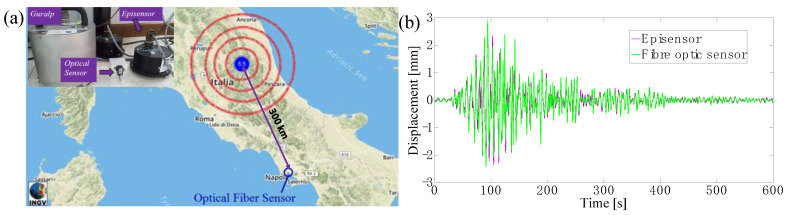
(**a**) Map of the Norcia’s earthquake epicenter. Inset: Picture of optical and reference seismic accelerometers installed at the INGV (**b**) Comparison between displacements recorded by Episensor and by the optical fiber sensor. (Reproduced with permission from ref. [70] (the figure is released under a CC-BY-SA license http://creativecommons.org/licenses/by/4.0/ (https://www.nature.com/articles/s41598-018-25082-8 (accessed on 27 April 2018)).

**Figure 16 sensors-23-03187-f016:**
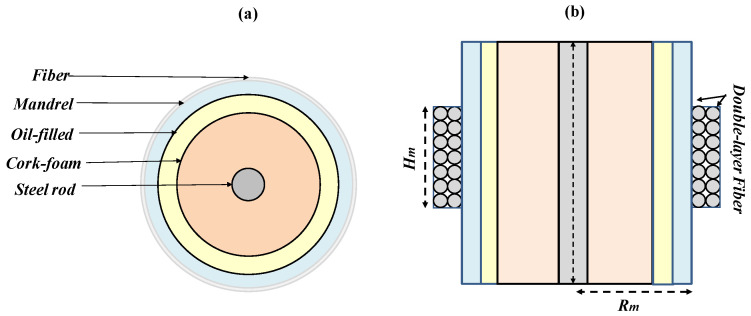
The composite FOH (**a**) horizontal cross-section and (**b**) lateral view. Reproduced with permission from ref [71], the figure is released under a Copyright Clearance Center’s RightsLink^®^ service).

**Figure 17 sensors-23-03187-f017:**
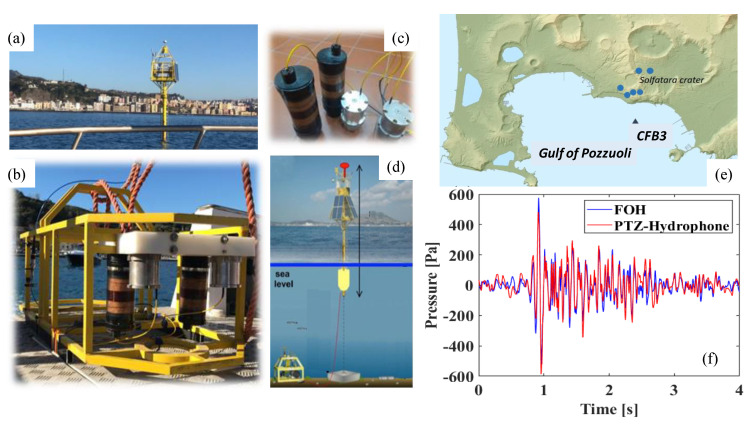
(**a**) Geodetic buoy of the INGV seismic monitoring network (**b**) Submarine module equipped with FOH sensors (**c**) Two FOHs (at left) and two dummy hydrophones (at right) (**d**) Sensing system installation scheme (**e**) Geographical Map of the Campi Flegrei caldera region. CFB3 is the buoy. Blue dots represent the epicenters of the main detected earthquakes (**f**) FOH (blue line) and PZT (red line) time response associated to one of the detected earthquakes (Reproduced with permission from ref [71], the figure is released under a Copyright Clearance Center’s RightsLink^®^ service).

**Figure 18 sensors-23-03187-f018:**
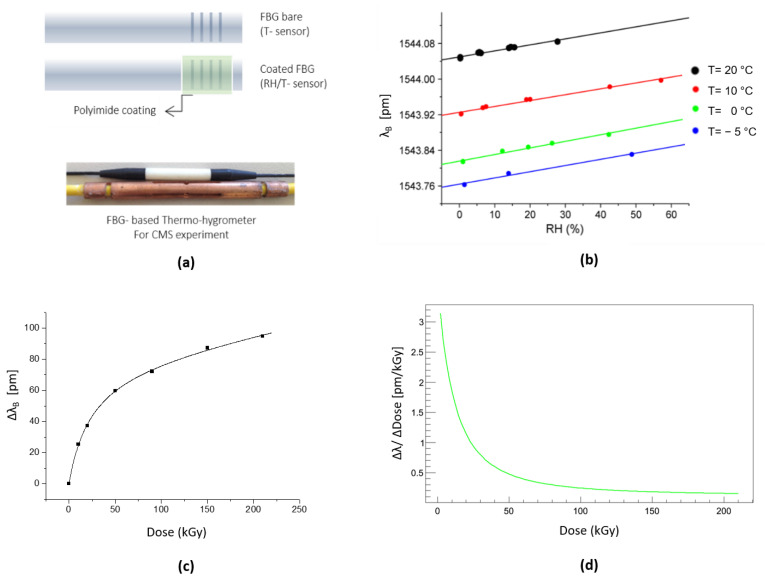
(**a**) FOS thermo-hygrometer based on FBG technology developed for CMS experiment; (**b**) RH characterization at different temperatures; (**c**) Radiation-induced wavelength shift as a function of the total adsorbed dose; (**d**) Radiation sensitivity (Readapted from Ref. [123]).

**Figure 19 sensors-23-03187-f019:**
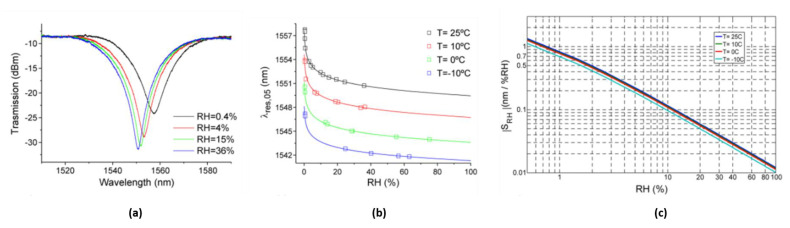
(**a**) Typical spectral RH response of TiO_2_-coated LPG at 25 °C. (**b**) RH characterization and (**c**) RH sensitivity at different temperatures (from Ref. [125]).

**Figure 20 sensors-23-03187-f020:**
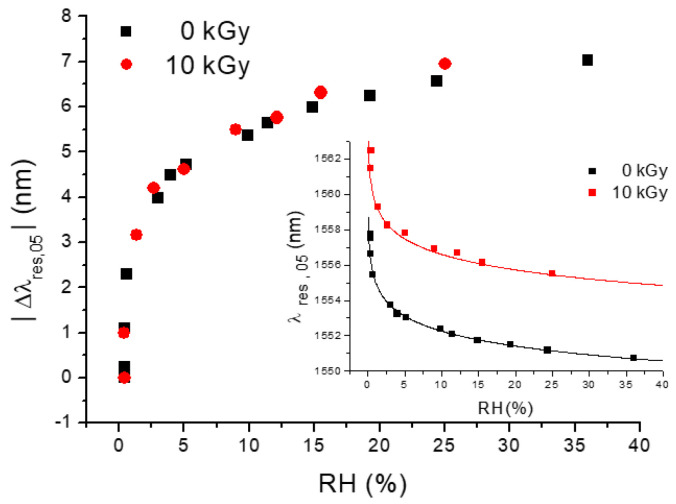
TiO_2_-coated LPG Rh characterization before (black points) and after (red points) 10 kGy dose of γ-ionizing radiation (from Ref. [126]).

**Figure 21 sensors-23-03187-f021:**
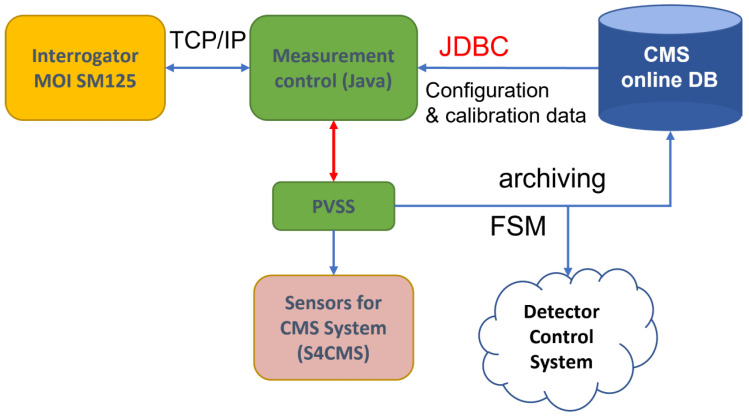
Integration of the FBG data taking system in the CMS Detector Control System.

**Figure 22 sensors-23-03187-f022:**
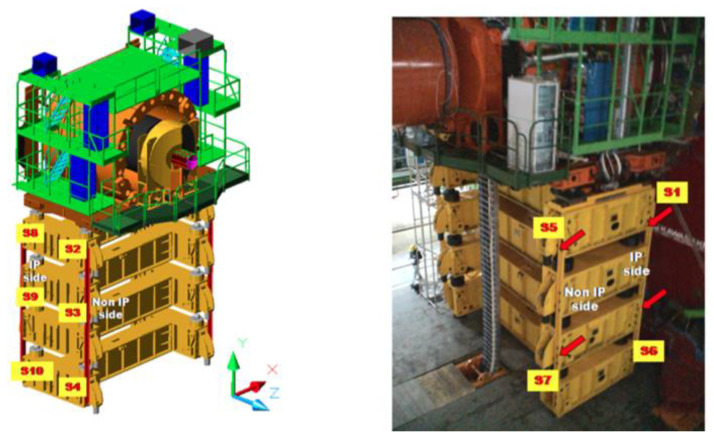
Layout of the FBG-based monitoring system installed on the HF raiser structure.

**Figure 23 sensors-23-03187-f023:**
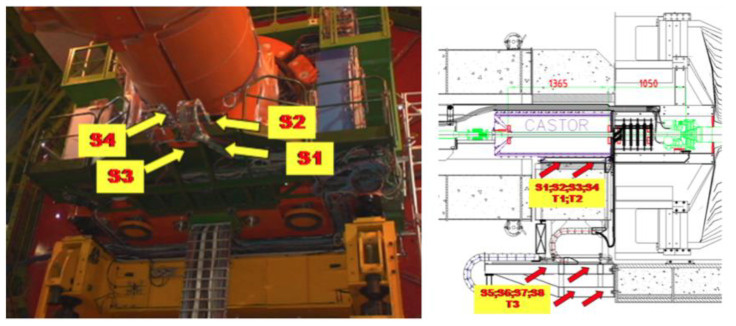
Layout of the FBG-based monitoring system installed on the CASTOR platform structure.

**Figure 24 sensors-23-03187-f024:**
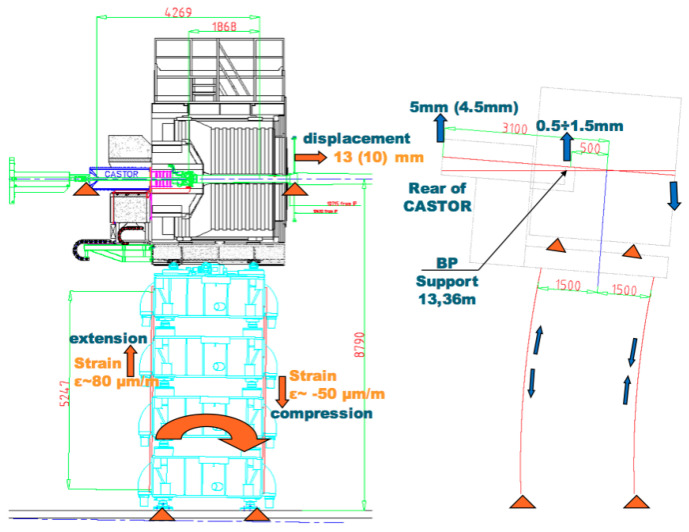
Morphing of the HF raiser and platform structure computed from the FBGs strain data.

**Figure 25 sensors-23-03187-f025:**
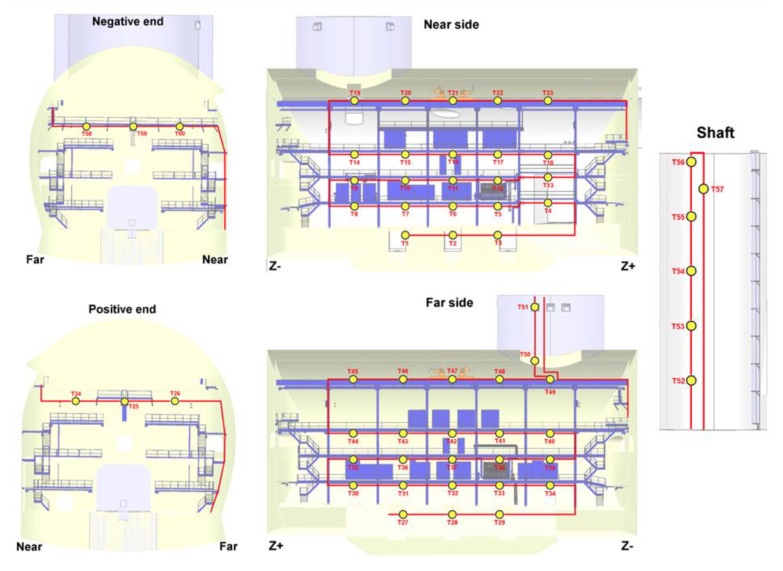
Layout of the CMS Experimental Underground Cavern (UXC) FBG-based temperature monitoring system. The positions of the FBG-based temperature sensors are denoted with “T” (from T1 to T57).

**Figure 26 sensors-23-03187-f026:**
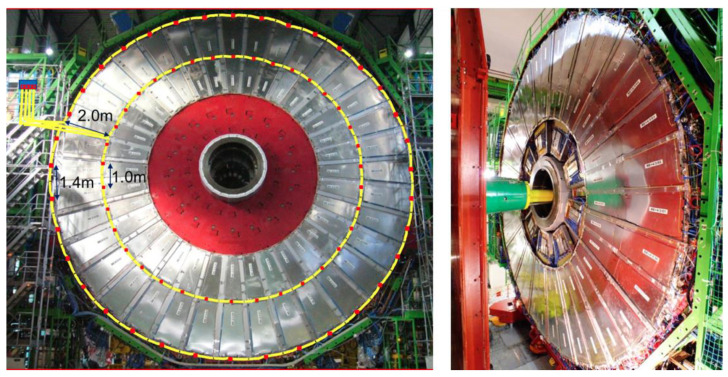
Layout of one of the six FBG temperature monitoring systems installed on the RPC endcap disks (**left**) and photo of the YE + 4 RPC disk with the FBGs array installed (**right**).

**Figure 27 sensors-23-03187-f027:**
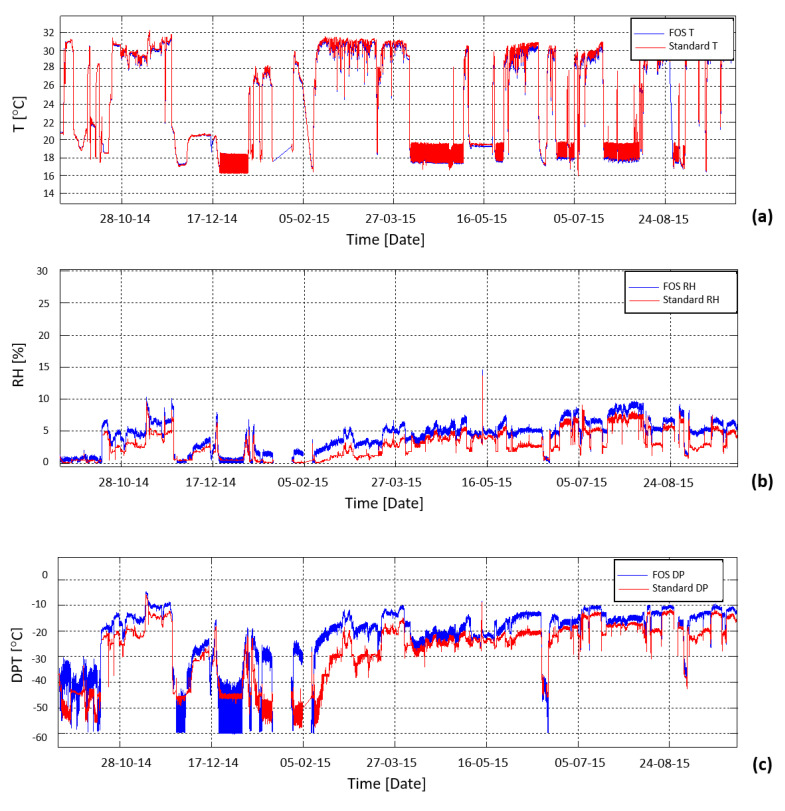
Temperature (**a**), relative humidity (**b**) and dew point (**c**) reconstruction from the FBG-based thermo-hygrometer installed in on the CMS Tracker end-flange. For comparison, the readings from the standard hygrometer installed in the same position are reported in red [125].

**Figure 29 sensors-23-03187-f029:**
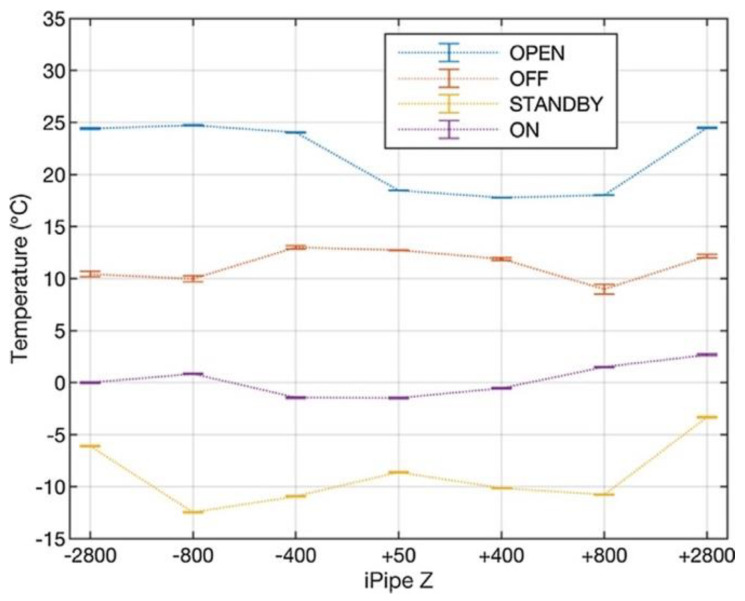
Thermal profile of the beam pipe relative to different operational status of the CMS detector, occurring in 2018 (adapted from Ref. [130]).

**Figure 30 sensors-23-03187-f030:**
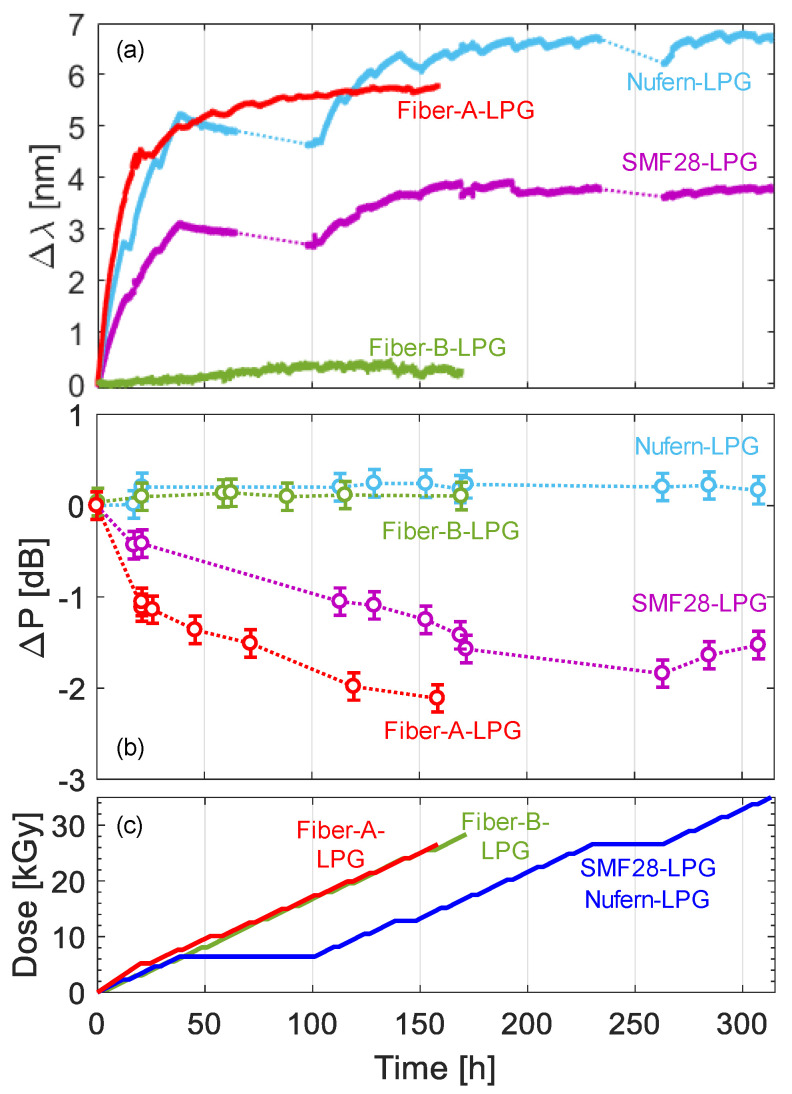
Real-time results of the gamma irradiation of LPGs arc-induced in different optical fibers: (**a**) resonance wavelength shift; (**b**) transmitted optical power changes; (**c**) irradiation profiles. (adapted from Ref. [147]).

**Figure 31 sensors-23-03187-f031:**
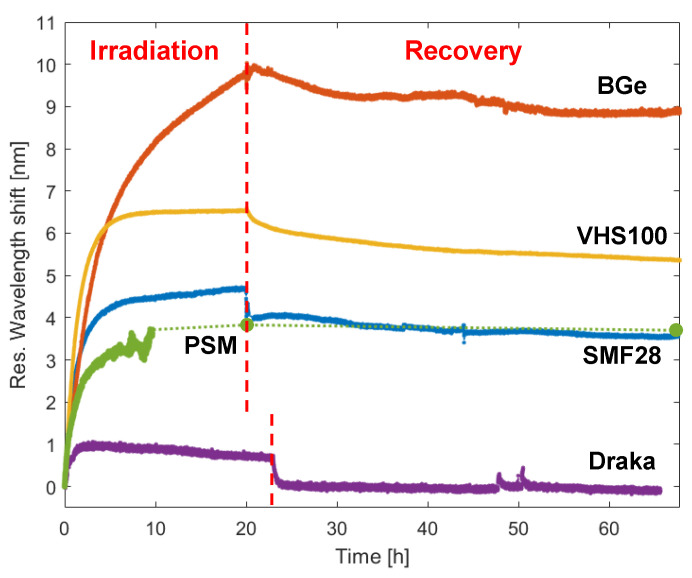
Real-time wavelength shift of LPGs written in several optical fibers, during gamma irradiation and subsequent recovery phase (adapted from Ref. [153]).

**Figure 32 sensors-23-03187-f032:**
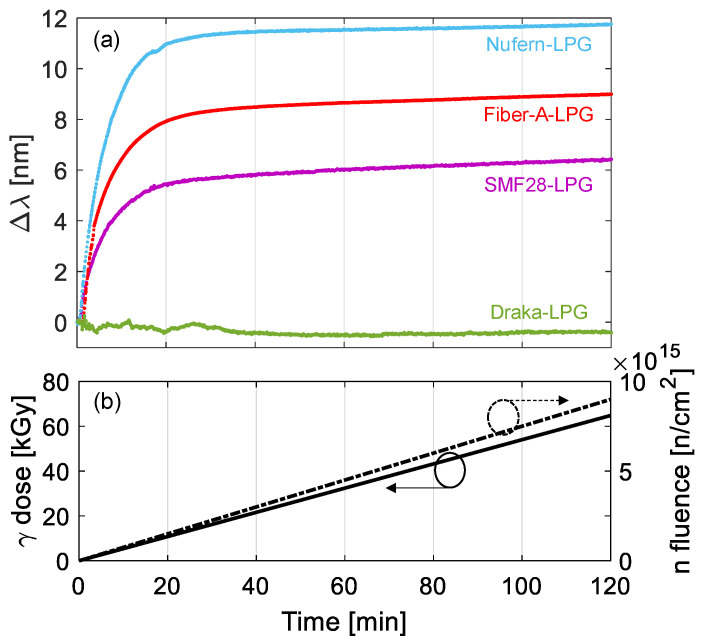
Real-time results of the mixed neutron-gamma irradiation of LPGs arc-induced in different optical fibers: (**a**) resonance wavelength shift; (**b**) irradiation profile (adapted from Ref. [154]).

**Figure 33 sensors-23-03187-f033:**
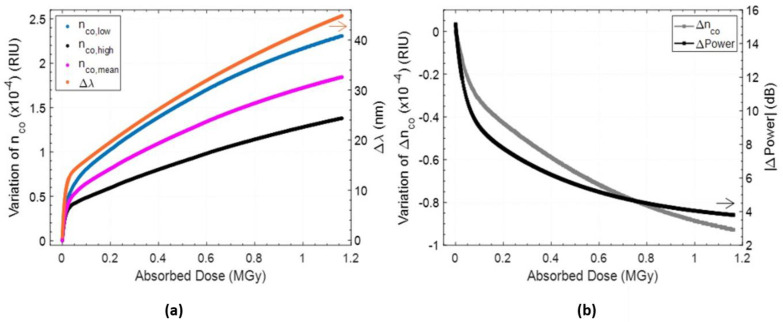
Variations of (**a**) n_co_ (on the left axis) and of Δλ (on the right axis), (**b**) Δn_co_ (on the left axis) and |ΔPower| (on the right axis) as a function of the absorbed dose. Adapted with permission from Ref. [156] (the figure is released under a Creative Commons CC BY license http://creativecommons.org/licenses/by/4.0/ (accessed on 12 March 2023), Copyright © 2018).

**Figure 34 sensors-23-03187-f034:**
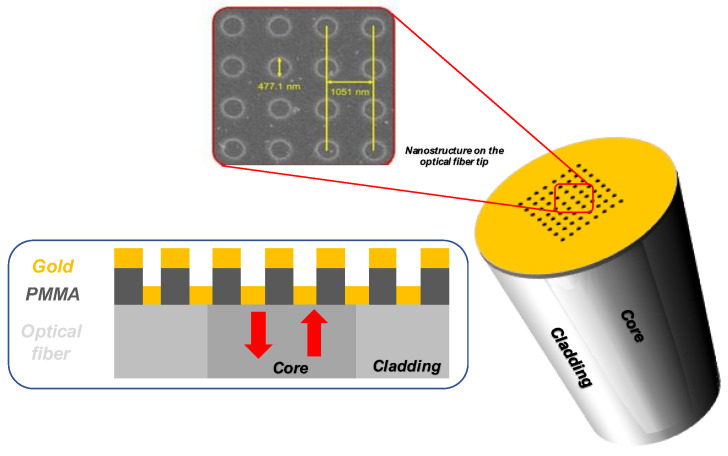
Cross section view and SEM image of the LOF resonator realized on the optical fiber tip. Adapted with permission from Ref. [159] (the figure is released under a Creative Commons CC BY license http://creativecommons.org/licenses/by/4.0/ (accessed on 12 March 2023), Copyright © 2018).

**Figure 35 sensors-23-03187-f035:**
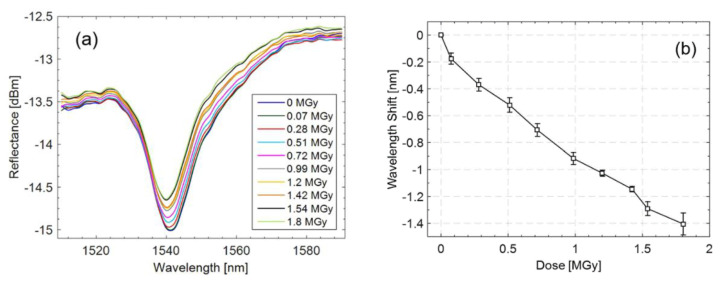
(**a**) LOF reflectance spectra acquired during the exposure to 23 GeV proton beam; (**b**) Calibration curve displaying the wavelength shift as a function of the dose. Reproduced with permission from Ref. [159] (the figure is released under a Creative Commons CC BY license http://creativecommons.org/licenses/by/4.0/ (accessed on 12 March 2023), Copyright © 2018).

**Figure 36 sensors-23-03187-f036:**
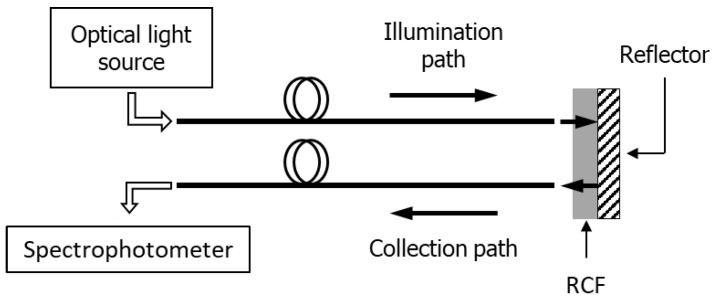
Typical schematic of an optical fiber based RCF reading method. Adapted with permission from Elsevier from Ref. [177] (the figure is released under a Copyright Clearance Center’s RightsLink^®^ service, Copyright © 2019).

**Figure 37 sensors-23-03187-f037:**
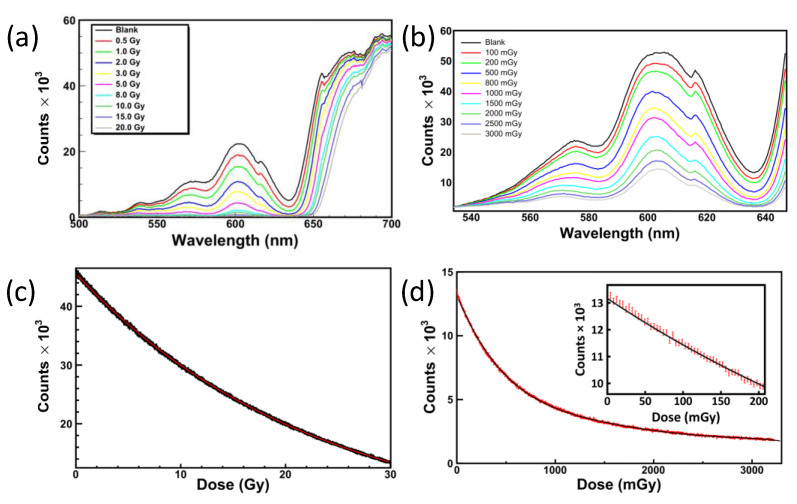
(**a**) Spectra of an EBT3 Gafchromic film exposed to ^60^Co γ-rays from the ISOF-CNR High Dose Rate irradiator at Bologna (Italy). (**b**) Spectra of an XR-QA2 Gafchromic film exposed to ^60^Co γ-rays from the Low Dose Rate irradiator. (**c**) Calibration of an EBT3 Gafchromic film exposed to ^60^Co-γ rays up to 30 Gy; the counts corresponding to the wavelength λ = 663 nm are used for the calibration. (**d**) Calibration of an XR-QA2 Gafchromic film exposed to ^60^Co-γ rays at mGy levels; the counts corresponding to the wavelength λ = 635 nm are used for the calibration; the zoom of the main plot in the range of dose [0–200 mGy] is shown in the inset. Reproduced with permission from Ref. [174] (the figure is released under a Creative Commons CC BY license http://creativecommons.org/licenses/by/4.0/ (accessed on 12 March 2023), Copyright © 2019).

**Figure 38 sensors-23-03187-f038:**
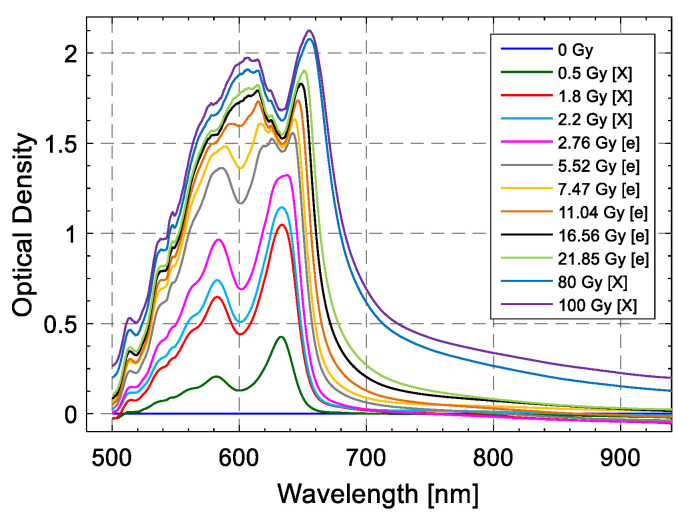
Average Optical Density spectra obtained upon five acquisitions from EBT3 film samples exposed to 1 MeV electron beam (referred as [e] in the legend) and to the 250 kV X-ray tube (referred as [X]). The null OD of the unexposed sample used as reference is also shown. Reprinted with permission from Elsevier from Ref. [177] (the figure is released under a Copyright Clearance Center’s RightsLink^®^ service, Copyright © 2019).

## Data Availability

Not applicable.

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
