# Peer review of "Innovative Photonic Sensors for Safety and Security, Part III: Environment, Agriculture and Soil Monitoring"

_sensors, 2023, doi:10.3390/s23063187_

Round 1

Reviewer 1 Report

The draft is a review article belong to a set of three companion papers.  It is the last one where authors focus on environmental monitoring by taking advantage of photonic technologies. It reports some work regarding soil water content measurement and landslide early warning, a new generation of seismic sensors useful in both terrestrial and under water contests is discussed, and a number of optical fiber sensors for use in radiation environments. The draft is written well but not organized well.  I would like to suggest the publication if the authors can carefully check the draft to present it better.

1 Many figures in the draft have not been organized well. For example, Fig. 1 and 2 can be combined together, because they are about the same thing. I also suggest combining Fig. 7 and 8, Fig. 19 and Fig. 20, Fig 21 and 22, Fig. 26 and 27, as well.

2 Some of Figures have low quality and it is difficult to read. The authors should use high quality figures. For example, Fig. 31 and Fig. 41(d), it is almost impossible to read the figure. I suggest the authors to check the figures again to make sure they are clear and not sloppy. For example, in Fig. 41, the sub figures should be the same size.

3 I would like to suggest a brief general introduction to connect this draft with the other two.

4 I suggest to rewrite the conclusion and future trends. It mentioned medical application, which is totally a spin off from the current draft.

5 There are some format errors, typo, and so on. For example, on page 19, there is a high light. At line 724, there is deleted number. Please double check them.

Author Response

 1 Many figures in the draft have not been organized well. For example, Fig. 1 and 2 can be combined together, because they are about the same thing. I also suggest combining Fig. 7 and 8, Fig. 19 and Fig. 20, Fig 21 and 22, Fig. 26 and 27, as well.

We thank the Reviewer for his/her suggestion. The figures 1-2, 7-8, 19-20, 21-22 have been combined. For figures 26 and 27, we think that it is better to keep them separated because each of them is composed by more sub-images and refer to different sensors’ installations.

2 Some of Figures have low quality and it is difficult to read. The authors should use high quality figures. For example, Fig. 31 and Fig. 41(d), it is almost impossible to read the figure. I suggest the authors to check the figures again to make sure they are clear and not sloppy. For example, in Fig. 41, the sub figures should be the same size.

We have improved the resolution of figures 31 and 41(d) and that of other figures whenever possible.

3 I would like to suggest a brief general introduction to connect this draft with the other two.

We have added a brief introduction linking this paper to the other two companions.

4 I suggest to rewrite the conclusion and future trends. It mentioned medical application, which is totally a spin off from the current draft.

We have modified the conclusions paragraph, in order to give more emphasis to the focus applications of this paper, while pointing out some advantages and disadvantages of the proposed solutions.

5 There are some format errors, typo, and so on. For example, on page 19, there is a high light. At line 724, there is deleted number. Please double check them.

We have double checked the manuscript and fixed the typos.

Reviewer 2 Report

The work is based of photonic sensor having application in pollution treatment. The conclusion may be made more clear. Rest are OK.

Author Response

The work is based of photonic sensor having application in pollution treatment. The conclusion may be made more clear. Rest are OK.

We thank the Reviewer for his/her positive comment. We have modified the conclusions paragraph in order to make it more clear to understand.

Reviewer 3 Report

This paper introduces many innovative applications of photonic sensors in the fields of environment, agriculture and soil monitoring, and grasps some schemes with research potential of optical fiber sensor in agriculture and soil monitoring, earthquake monitoring and high radiation scenarios. It can be seen that the author has grasped the current development context and research status of optical fiber sensor. The article is clear and easy to understand, and the content of the arrangement is also detailed, including not only the introduction of the principle but also the experimental analysis, which can better increase the credibility of the schemes.

Some comments:

-The title of the article is about “Innovative photonic sensors”, but the content is only the "Optical Fiber Sensor" section of "Photonic Sensor".

-Figure 3 (a): Only the red box is used to select different VWC values, which is not clear in the part of the figure. It is recommended to clearly mark when and how the VWC changes in the graph.

-Figure 10 (b): The meaning represented by the position of the x-axis in the figure should be clearly explained.

-Figure 9 (b):ΔP is proposed directly without a clear definition.

-Figure 15 (a): The model is not clear and intuitive, for example, appropriate tagging of each part can be added so that the reader can easier to understand.

-Number list in the conclusion is not suggested, which can be replaced with bullet format.

-Consistence of the references should be improved, for example, abbreviation or not for the journal name, and volume, issue and page number.

In addition, I suggest to increase the analysis of the advantages and disadvantages of each scheme, the analysis of applicable conditions, etc., or analyze why you choose these schemes.

Author Response

1) The title of the article is about “Innovative photonic sensors”, but the content is only the "Optical Fiber Sensor" section of "Photonic Sensor".

We thank the Reviewer. We agree with this comment, however we should remark that this is the third of a series of three companion papers, which are not restricted to optical fiber sensors. Therefore, we prefer to keep the expression “Innovative photonic sensors” in the title, in order to keep this paper linked to the other two companion papers published on the same special issue.

2) Figure 3 (a): Only the red box is used to select different VWC values, which is not clear in the part of the figure. It is recommended to clearly mark when and how the VWC changes in the graph.

We have changed the figure, which now clearly indicates the VWC values corresponding to the different portions of the acquired data.

3) Figure 10 (b): The meaning represented by the position of the x-axis in the figure should be clearly explained.

The meaning of the x-axis in Fig. 10(b) has been clarified in the revised manuscript.

4) Figure 9 (b):ΔP is proposed directly without a clear definition.

In the revised manuscript, the quantity ΔP has been defined in the caption of the figure.

 5) Figure 15 (a): The model is not clear and intuitive, for example, appropriate tagging of each part can be added so that the reader can easier to understand.

The figure has been modified adding the meaning of its different parts.

 6) Number list in the conclusion is not suggested, which can be replaced with bullet format.

We have modified the conclusions avoiding the use of a numbered list.

7) Consistence of the references should be improved, for example, abbreviation or not for the journal name, and volume, issue and page number.

We have revised the references, using the format specified by the journal template.

8) In addition, I suggest to increase the analysis of the advantages and disadvantages of each scheme, the analysis of applicable conditions, etc., or analyze why you choose these schemes.

We thank the Reviewer for his/her comments. We have added some details in the conclusions paragraph. However, we also wish to stress out that, in this work - the third one of a set of three companion papers - we focus on the use of some innovative photonic sensors in environmental, radiation and soil monitoring. Due to the large variety of applications, it is not easy to compare the various schemes presented in this paper, as they refer to different contexts. Nonetheless, we appreciate the reviewer suggestion and evaluate the opportunity of a future publication where various photonic sensor solutions are compared within each application field.

Round 2

Reviewer 3 Report

The authors have made the revisions accordingly.